# Efficient Temporal Action Segmentation via Boundary-aware Query Voting

**Peiyao Wang**[1]     **Yuewei Lin**[2]     **Erik Blasch**[3]     **Jie Wei**[4]     **Haibin Ling**[1*]

[1]Stony Brook University, [2]Brookhaven National Laboratory,
[3]Air Force Research Laboratory, [4]The City College of New York
{peiyaowang, hling}@cs.stonybrook.edu, ywlin@bnl.gov,
erik.blasch@gmail.com, jwei@ccny.cuny.edu

## Abstract

Although the performance of Temporal Action Segmentation (TAS) has been improved in recent years, achieving promising results often comes with a high computational cost due to dense inputs, complex model structures, and resource-intensive post-processing requirements. To improve the efficiency while keeping the high performance, we present a novel perspective centered on per-segment classification. By harnessing the capabilities of Transformers, we tokenize each video segment as an instance token, endowed with intrinsic instance segmentation. To realize efficient action segmentation, we introduce BaFormer, a boundary-aware Transformer network. It employs instance queries for instance segmentation and a global query for class-agnostic boundary prediction, yielding continuous segment proposals. During inference, BaFormer employs a simple yet effective voting strategy to classify boundary-wise segments based on instance segmentation. Remarkably, as a single-stage approach, BaFormer significantly reduces the computational costs, utilizing only ∼**6%** of the running time compared to the state-of-the-art method DiffAct, while producing better or comparable accuracy over several popular benchmarks. The code for this project is publicly available at https://github.com/peiyao-w/BaFormer.

## 1 Introduction

Temporal Action Segmentation (TAS) [28, 43, 30, 29, 21, 32], a notable endeavor in the study of untrimmed videos, aims to allocate an action label to each frame, enabling the detailed analysis of complex activities by identifying specific actions within long-form videos. It has extensive applications in surveillance [11, 12], sports analytics [20], and human-computer interaction [38]. Despite its critical importance, existing TAS models often grapple with high computational costs and lengthy inference times, limiting their applicability in real-time and resource-constrained scenarios.

The inefficiency of TAS arises from two primary factors:1) Long-form Input: Untrimmed videos in TAS often include tens of thousands of frames containing multiple action segments that ideally should be sparse in terms of action semantics. However, most approaches [15, 32, 21, 43] maintain frame-wise processing throughout the model to preserve dense information, aiming to enhance frame-wise prediction accuracy yet at the cost of efficiency. Conversely, the straightforward downsampling method [28] fails to achieve the desired accuracy for dense predictions. 2) Heavy Model Structure: The adoption of multi-stage refinement models and various time-consuming post-processing techniques further contribute to the slow runtime of TAS models. For instance, although a previous

---

*Corresponding authors

38th Conference on Neural Information Processing Systems (NeurIPS 2024).

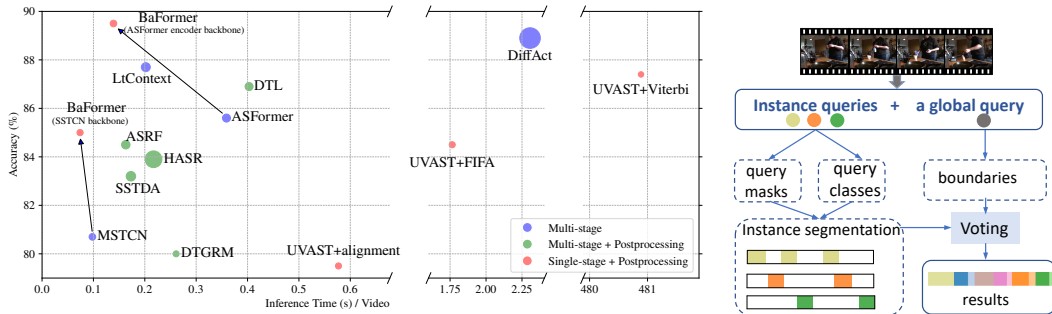

Figure 1: Accuray vs. inference time on 50Salads. The bubble size represents the FLOPs in inference. Under different backbones, BaFormer enjoys the benefit of boundary-aware query voting with less running time and improved accuracy.

Figure 2: An efficient pipeline developed by our proposed BaFormer.

method [3] employs a single-stage model with transcripts to mitigate the heavy model structure, the algorithms used during post-processing remain time-intensive, as shown in Fig. 1.

**Our objective is to transform the long-form video into a sparse representation, thereby reducing the temporal dimension processed by the model. Additionally, we aim to employ a single-stage model coupled with an appropriate post-processing method to minimize the running time.** Benefiting from the query-based model, which allows information to be compressed into a limited number of queries, segments can be effectively represented as queries. By enabling each query to predict the corresponding class, start, and end, action segmentation can be seamlessly reframed as an action detection task. However, this significant compression of the long-form input adversely impacts the accuracy of dense predictions, as noted in [3]. To address the limitation, we draw inspiration from recent advances in mask classification-based image segmentation, particularly MaskFormer [10]. In TAS, rather than predicting frame-wise results directly, we propose using a set of queries to represent segments by decoupling their binary masks and their classes. The decoupling approach effectively conveys sparse information through limited queries, while preserving dense information via binary masks. Furthermore, the continuous nature of features in the temporal dimension introduces discontinuities in learning frame-wise masks. To address the issue of discontinuity, we introduce an additional query to learn boundaries, capturing global information with minimal computational overhead. The additional query should also facilitate efficient post-processing.

**Our solution**, depicted in Figure 2, involves a parallel boundary-wise segment proposal generation to yield compact and continuous segment proposals. By classifying these segment proposals via multiple query voting, our method, named **BaFormer** (Boundary-aware Transformer), realizes efficient TAS with high performance. Experiments conducted on three popular TAS datasets validate the effectiveness of our method. Specifically, on the 50Salads dataset: compared with the baseline ASFormer, our approach improves accuracy by 3.9%, F1@50 by 7.9%, and edit score by 4.6%. Compared to the state-of-the-art performer DiffAct [32], our approach delivers competitive results while requiring only **10%** of the FLOPs and **6%** running time. Moreover, our query-based voting mechanism significantly reduces inference time to **24%** of that required by the single-stage model UVAST [3]. The contributions of our work are threefold:

1. A query-based Transformer is introduced to convert TAS from dense per-frame classification into a sparse per-query alternative, substantially reducing computational overhead.

2. The simple yet effective boundary-aware query voting ensures accurate and coherent action segmentation.

3. Experimental evidence demonstrates our model's superiority in efficiency over existing state-of-the-art TAS methods, without sacrificing effectiveness.

## 2 Related Work

**Temporal Action Segmentation.** In temporal action segmentation, many methods [15, 30, 7, 1, 41, 21, 45, 6, 44, 32, 3] have been developed to address the challenge of over-segmentation. Notably,

MSTCN [15], ASFormer [45], and DiffAct [32] utilize repeating modules and multi-loss supervision to enhance long-range dependency capture and refine preliminary predictions. In addition, several methods [7, 1, 41, 21, 6, 44, 18] incorporate post-processing techniques that integrate priors, such as boundaries [43] and transcripts [3], to further refine per-frame predictions. Additionally, other methods[43, 21, 6] opt for a parallel approach to frame-wise prediction and constraint learning within end-to-end models. For instance, BCN [43] and ASRF [21] introduce boundary prediction as an auxiliary branch to enhance results, while UARL [6] leverages certainty learning to exploit transitional expressions. Despite these innovative efforts improving the performance, the reliance on multistage networks as the backbone structure leads to increased computation costs. Building on recent advancements, UVAST [3] introduces a query-based network [40] to predict transcripts, transitioning from a multi-stage to a single-stage network for frame-wise prediction, which results in a reduction of FLOPs compared to earlier methods. However, post-processing with the Viterbi algorithm [26, 36] incurs significant time during inference, detracting from its suitability for real-time applications.

**Query-based model for Image Detection and Segmentation.** Benefiting from Transformer architectures, query-based models have recently revolutionized object detection [4, 34, 13, 37, 33, 46, 8], which utilizes a Transformer network to predict bounding boxes and their corresponding classes. This architecture can be readily adapted for image segmentation as well. MaskFormer [10] and Mask2Former [9] offer a direct application, that predicts mask classifications and their corresponding binary masks, transforming the task from pixel-wise dense prediction to mask classification. In recent advancements, SAM (Segment Anything Model) [24] emerges as a powerful solution for segmenting generic objects. Integrating query-based architectures has paved the way for more effective and versatile segmentation solutions.

**Our method** is different from previous TAS ones with the use of the query-based model. UVAST [3], yields a query-based transcript used for the potentially heavy post-processing, is mostly related to ours. By contrast, our approach BaFormer utilizes the query-based module in conjunction with a single-stage frame-wise module to decouple frame-wise results into query masks and classifications like that in image segmentation via mask classification, which is the first for TAS. The decoupling allows our method to facilitate efficient inference via query-based majority voting, leveraging boundary information.

## 3 BaFormer

The overview of BaFormer is shown in Fig. 3, consisting of a frame-wise encoder-decoder, a Transformer decoder, output heads, and inference processing. The input is a video $\mathbf{V} \in \mathbb{R}^{T \times 3 \times H \times W}$ comprising $T$ frames of resolution $H \times W$, and the final output is frame-wise segment results.

- In the frame-wise encoder-decoder module, the frame encoder first extracts video feature maps $\mathbf{F}^e \in \mathbb{R}^{T \times C_0}$ from the input video $\mathbf{V}$, which are then fed to frame decoder with $L$ layers followed by a linear layer, resulting in $\mathbf{F}^d \in \mathbb{R}^{T \times C}$. Additionally, we collect output features from $L$ frame decoder layers as $\mathbf{F} = \{\mathbf{f}_l \in \mathbb{R}^{T \times C}\}_{l=1}^{L}$.

- Next, the Transformer decoder, comprising $L$ layers, processes $M$ trainable instance queries $\mathbf{Q}_0 \in \mathbb{R}^{M \times C}$. It takes features $\mathbf{F}$ and masks $\mathcal{M} = \{\mathbf{P}_l^m \in \mathbb{R}^{M \times T}\}_{l=0}^{L-1}$ as inputs. $\mathbf{P}_l^m = \varphi^m(\mathbf{Q}_l, \mathbf{F}^d)$ corresponds to the mask prediction probability estimated by the mask prediction head $\varphi^m$. $\mathbf{Q}_l$ is instance query embedding of the $l^{\text{th}}$ Transformer layer. As shown in Fig. 4(a), the $l^{\text{th}}$ layer utilizes $\mathbf{Q}_{l-1}, \mathbf{P}_{l-1}^m, \mathbf{f}_l$ to generate $\mathbf{Q}_l$.

- In the output heads, each instance query embedding $\mathbf{Q}_l$ is processed alongside frame embeddings $\mathbf{F}^d$ by three distinct heads, which calculate the probabilities for query class ($\mathbf{P}_l^c$), masks ($\mathbf{P}_l^m$), and boundaries ($\mathbf{P}_l^b$), respectively. Fig. 3 specifically illustrates the processing of the final layer's embeddings by the output heads.

- Finally, during inference, the predictions are aggregate from the final Transformer layer, including query class probability $\mathbf{P}_L^c$, masks prediction $\mathbf{P}_L^m$, and boundary predictions $\mathbf{P}_L^b$. And aboundary-aware query voting is employed to finalize the results.

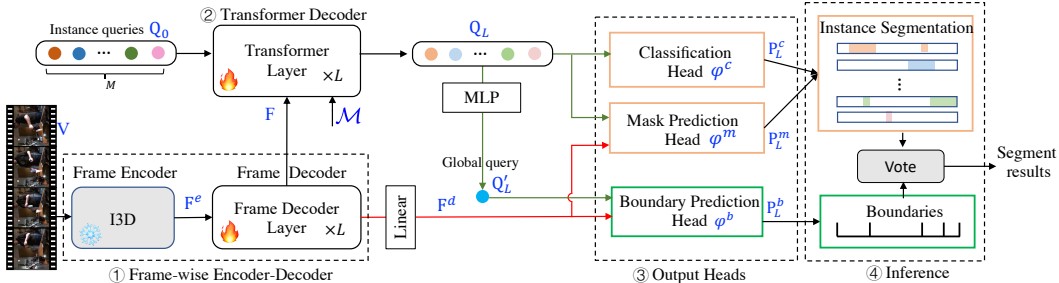

Figure 3: Overview of BaFormer architecture. It predicts query classes and masks, along with boundaries from output heads. Although each layer in the Transformer decoder holds three heads, we illustrate the three heads in the last layer for simplicity.

## 3.1 Frame-wise Encoder-Decoder

The frame-wise encoder-decoder module is designed to preserve dense information essential for model's functionality. Following previous research [15, 45, 32], we employ the I3D [5] encoder with frozen parameters to transfer video input $\mathbf{V} \in \mathbb{R}^{T \times 3 \times H \times W}$ into video features $\mathbf{F}^e \in \mathbb{R}^{T \times C_0}$, setting $C_0$ to 2048. Subsequently, we select the ASFormer encoder [45], equipped with $L$ layers, as the frame-wise decoder. This decoder processes $\mathbf{F}^e$ to yield frame-wise embeddings $\mathbf{F}^d \in \mathbb{R}^{T \times C}$, enriching the model with dense information crucial for query masks and boundary predictions.

## 3.2 Transformer Decoder

The Transformer decoder aims to compress video sequences into sparse representations via queries to improve model efficiency. It stacks $L$ Transformer layers $\{\Omega_l\}_{l=1}^L$, as illustrated in Fig. 4(a). The output of the $l^{\text{th}}$ layer $\mathbf{Q}_l \in \mathbb{R}^{M \times C}$ can be generated according to:

$$\mathbf{Q}_l = \Omega_l(\mathbf{Q}_{l-1}, \mathbf{P}_{l-1}^m, \mathbf{f}_l), \quad l = 1, 2, ..., L \tag{1}$$

where $\mathbf{f}_l \in \mathbb{R}^{T \times C}$ represents the frame-wise features from the $l^{\text{th}}$ frame decoder layer, and $\mathbf{P}_{l-1}^m = \varphi^m(\mathbf{Q}_{l-1}, \mathbf{F}^d) \in \mathbb{R}^{M \times T}$ is the output of mask prediction head $\varphi^m$ with input $\mathbf{Q}_{l-1}$ and $\mathbf{F}^d$. The detailed computation of $\varphi^m$ will be specified in the output heads section.

The final output of the Transformer decoder, $\mathbf{Q}_L$, is then directed to the output heads for the ultimate prediction. Additionally, the intermediate outputs $\mathbf{Q}_l$ (for $l < L$) are also fed into output heads, serving for auxiliary outputs, which are further discussed in output heads and loss sections.

The specific structure of each Transformer layer $\Omega_l$ is illustrated in Fig. 4(b). Each layer consists of three sub-layers with normalization and residual connection: (1) a Mask-Attention (MA) layer, (2) a Self-Attention (SA) layer, and (3) a Feed-Forward Fetwork (FFN). The SA and FFN are the same as those in the vanilla Transformer [40]. The MA layer is a variant from Mask2Former [9], where the thresholding of mask prediction is changed from hard into soft, which suits action segmentation within a temporally continuous feature space. Specifically, in the $l^{\text{th}}$ Transformer layer, mask-attention layer in Fig. 4(b) can be formulated as:

$$\hat{\mathbf{Q}}_l = \text{Linear}(\mathbf{Q}_{l-1}), \quad \hat{\mathbf{K}}_l = \text{Linear}(\mathbf{f}_l), \quad \hat{\mathbf{V}}_l = \text{Linear}(\mathbf{f}_l),$$
$$\mathbf{X}_l = \text{Linear}\Big(\text{softmax}\big(\frac{\mathbf{P}_{l-1}^m}{\sqrt{C}} \odot \hat{\mathbf{Q}}_l \hat{\mathbf{K}}_l^T\big)\hat{\mathbf{V}}_l\Big) + \mathbf{Q}_{l-1}, \tag{2}$$

where $\odot$ is the Hadamard product. The mask $\mathbf{P}_{l-1}^m$ is a probability matrix, with each element within the interval $[0, 1]$. This structure enables the queries to concentrate more effectively on the principal information of the frame sequence.

## 3.3 Output Heads

The output heads are designed to generate key elements required by the inference module, including query classes, query masks, and class-agnostic boundaries. Given $\{\mathbf{Q}_l\}_{l=1}^L$, there will be $L$ groups of output heads. Each $\mathbf{Q}_l$ is processed through the $l^{\text{th}}$ group of output heads to yield prediction. For

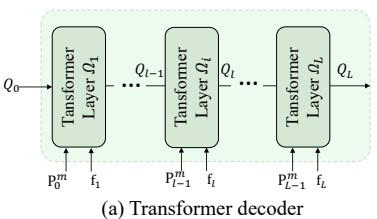
(a) Transformer decoder

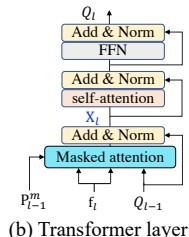
(b) Transformer layer

Figure 4: Details of Transformer decoder. (a) Transformer decoder stacks $L$ Transformer layers. (b) Each Transformer layer consists of a masked attention, self-attention, and a feedforward network with residual connections and normalization.

notation conciseness, we omit the subscripts of the inputs $\mathbf{Q}_l$, the output heads, and the predictions in this section.

Each group of output heads comprises three learnable modules: a classification head $\varphi^c$, a mask prediction head $\varphi^m$, and a boundary prediction head $\varphi^b$. They will produce query class probability $\mathbf{P}^c \in \mathbb{R}^{M \times (K+1)}$, query masks $\mathbf{P}^m \in \mathbb{R}^{M \times T}$, and boundary probability $\mathbf{P}^b \in \mathbb{R}^T$, respectively. They can be formulated as:

$$\mathbf{P}^c = \varphi^c(\mathbf{Q}) = \text{softmax}(\text{Linear}(\mathbf{Q})), \quad \mathbf{P}^m = \varphi^c(\mathbf{Q}, \mathbf{F}^d) = \text{sigmoid}(\text{MLP}(\mathbf{Q})\mathbf{F}^{d\mathsf{T}}). \quad (3)$$

The Multi-Layer Perceptron (MLP) consists of three hidden layers. According to the $M$ queries, we can denote output as $\mathbf{P}^c = \{\mathbf{p}_i^c\}_{i=1}^M$, $\mathbf{P}^m = \{\mathbf{p}_i^m\}_{i=1}^M$. The predicted class-mask pair $(\mathbf{p}_i^c, \mathbf{p}_i^m)$ corresponds to output of the $i^{\text{th}}$ query.

Before the instance query embeddings $\mathbf{Q}$ are fed into the boundary head, they are aggregated across the query dimension into a global query by another MLP with three layers, followed by a linear layer, resulting in $\mathbf{Q}' = \text{Linear}(\text{MLP}(\mathbf{Q})^\mathsf{T})^\mathsf{T} \in \mathbb{R}^C$. The global query encompasses comprehensive video information, thereby facilitating a global boundary prediction. Then the boundary head can be formulated as:

$$\mathbf{P}^b = \varphi^b(\mathbf{Q}', \mathbf{F}^d) = \text{sigmoid}(\mathbf{Q}'\mathbf{F}^{d\mathsf{T}}). \quad (4)$$

Moreover, outputs of previous Transformer decoder layers $\mathbf{Q}_l (l < L)$ are also fed into outputs heads through Equations (3) and (4) to obtain auxiliary outputs $\mathbf{P}_l^c, \mathbf{P}_l^m$, and $\mathbf{P}_l^b$, serving as intermediate supervision.

### 3.4 Matching Strategies and Loss Function

Training the model requires ground truth supervision for query classes, masks, and global boundaries. For boundary detection, we represent ground-truth boundaries as $\mathbf{y}^b \in \mathbb{R}^T$, a heatmap generated by applying a Gaussian distribution to the binary boundary mask. For query class and mask ground truths, we define ground-truth segments as class-mask pairs $\{(y_i^c, \mathbf{y}_i^m)\}_{i=1}^N$, where $y_i^c \in \mathbb{R}$ indicates the action class, $\mathbf{y}_i^m \in \mathbb{R}^T$ is the corresponding binary mask, and $N$ represents the total number of such segments in the video.

The predicted outputs for classes and masks, denoted as $\mathbf{P}^c$ and $\mathbf{P}^m$, are organized into class-mask pairs $\{(\mathbf{p}_i^c, \mathbf{p}_i^m)\}_{i=1}^M$. $\mathbf{p}_i^c \in \mathbb{R}^{K+1}$ and $\mathbf{p}_i^m \in \mathbb{R}^T$ represent the $i^{\text{th}}$ query's class and mask probability prediction, respectively, and $K$ is the number of action class in the dataset.

Before loss computation, a crucial step is to match ground-truth data with query predictions. Depending on the ground-truth segment type, we apply different matching strategies, as shown in Fig. 5: (1) Ordered Class Matching: instance queries are sequentially aligned with the class order, ensuring a direct match. (2) Transcript Matching: instance queries are sequentially aligned with transcript ground truths by action order, with any excess query predictions aligned to a no-action label. (3) Instance Matching: instance queries are matched with instance ground truths by Hungarian matching algorithm [27], with extra query predictions assigned to a no-action label. The first two strategies employ fixed matching based on query order, whereas the third strategy utilizes dynamic matching, requiring the matching cost from query class matching and mask matching. We define the cost when matching a pair of mask $\mathbf{a} \in \mathbb{R}^T$ and $\mathbf{b} \in \mathbb{R}^T$ as:

$$\mathcal{L}_{\text{mask}}(\mathbf{a}, \mathbf{b}) = \lambda_{\text{focal}}\mathcal{L}_{\text{focal}}(\mathbf{a}, \mathbf{b}) + \lambda_{\text{dice}}\mathcal{L}_{\text{dice}}(\mathbf{a}, \mathbf{b}), \quad (5)$$

where the $\lambda_{\text{focal}}$ and $\lambda_{\text{dice}}$ are the loss weight of focal loss [31] and dice loss [35]. Defined as $\mathbf{p}_{(i,j)}^c$, the probability value in the $i^{\text{th}}$ query class prediction $\mathbf{p}_i^c$ is indexed by the $j^{\text{th}}$ class. So the cost

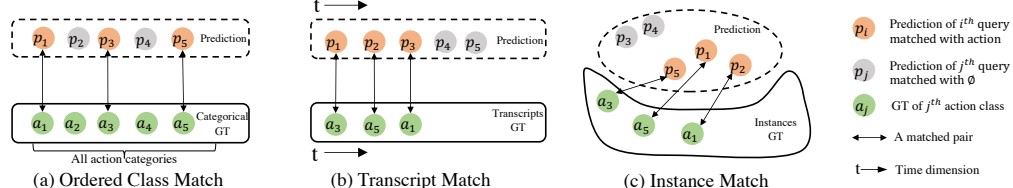

(a) Ordered Class Match      (b) Transcript Match      (c) Instance Match

Figure 5: Different matching strategies. Given an example video including ordered action $[a_3, a_5, a_1]$ from a dataset with all action classes $\{a_i\}_{i=1}^5$, (a) and (b) are fixed matching, while (c) is dynamic matching.

between the $i^{\text{th}}$ query prediction $(\mathbf{p}_i^c, \mathbf{p}_i^m)$ and the $j^{\text{th}}$ ground truth $(y_j^c, \mathbf{y}_j^m)$ in Hungarian matching is:

$$\text{Cost}_{ij} = -\log \mathbf{p}_{(i,y_j^c)}^c + \mathcal{L}_{\text{mask}}(\mathbf{p}_i^m, \mathbf{y}_j^m). \tag{6}$$

Experimental results reveal that instance matching outperforms others. Training losses for ordered class and transcript matching detailed in the supplemental material. When utilizing instance matching, the training loss for a video is defined as follows:

$$\mathcal{L} = -\sum_{i=1}^{M} \log \mathbf{p}_{(i,y_{\sigma(i)}^c)}^c + \sum_{j=1}^{N} \mathcal{L}_{\text{mask}}(\mathbf{p}_{\delta(j)}^m, \mathbf{y}_j^m) - \sum_{t=1}^{T} y_t^b \log p_t^b, \tag{7}$$

where $\sigma(i)$ is the optimal ground truth index aligned with the $i^{\text{th}}$ query prediction, $\delta(j)$ denotes the optimal prediction index corresponding to the $j^{\text{th}}$ ground truth, $y_t^b$ is the $t^{\text{th}}$ element of $\mathbf{y}^b$, and $p_t^b$ is the $t^{\text{th}}$ element of $\mathbf{P}^b$.

### 3.5 Inference

During inference, a set of $M$ pairs of query class-mask probability $\{(\mathbf{p}_i^c, \mathbf{p}_i^m)\}_{i=1}^M$ are obtained, along with a boundary probability vector $\mathbf{P}^b \in \mathbb{R}^T$, from the $L^{\text{th}}$ output heads. We apply a dedicated algorithm outlined in Algorithm 1 to derive the ultimate segmentation outcomes. Initially, boundary prediction produces class-agnostic segment proposals. In each segment, each query mask within the segment is summed up and we identify the majority-contributing query for the segment. Ultimately, the class of this predominant query is assigned as the class for the segment. The algorithm is based on the observation that for each segment proposal span, all the queries may be activated in the mask prediction but the query corresponding to the correct category always dominates.

---

**Algorithm 1:** Boundary-aware Query Voting

---

**Input:** Probability of query class–mask pairs: $\{(\mathbf{p}_i^c, \mathbf{p}_i^m)\}_{i=1}^M$, where
$\mathbf{p}_i^c \in \mathbb{R}^{K+1}, \mathbf{p}_i^m \in \mathbb{R}^T$; Boundary probability: $\mathbf{P}^b = \{p_t^b\}_{t=1}^T$, where
$p_t^b \in \mathbb{R}$ is the boundary probability in the $t^{\text{th}}$ frame.
**Output:** Frame-wise segmentation: $\mathbf{S} \in \mathbb{R}^T$.

1   Initialize $\mathbf{S} \in \mathbb{R}^T$ with all zeros
2   $\mathbf{C} \leftarrow \{\text{cls}_i | \text{cls}_i = \text{argmax}(\mathbf{p}_i^c[: K])\}_{i=1}^M$
3   $\mathbf{B} = \{b_i\}_{i=1}^{N_b} \leftarrow \text{sort}(\{1, T\} \bigcup \{t | (p_t^b > p_{t-1}^b) \& (p_t^b > p_{t+1}^b), 1 < t < T\})$
4   **for** $i = 1, 2, ..., N_b - 1$ **do**
5      **for** $j = 1, 2, ..., M$ **do**
6        $w_{ij} = \sum \mathbf{p}_j^m[b_i : b_{i+1}]$
7      **end**
8      $k = \text{argmax}_j(\{w_{ij}\}_{j=1}^M)$
9      $\mathbf{S}[b_i : b_{i+1}] = \text{cls}_k$
10   **end**

---

## 4 Experiments

### 4.1 Setup

**Dataset and Evaluation Metric.** We use three challenging datasets, GTEA [16], 50Salads [39] and Breakfast [25], where 50Salads presents the longest videos while GTEA includes the shortest

| Match | #Q | FLOP (G) | Time (s) | Para (M) | F1 @{10, 25, 50} | | | Edit | Acc. |
|---|---|---|---|---|---|---|---|---|---|
| Ordered Class | 19 | 3.74 | 0.136 | 1.49 | 88.1 | 87.0 | 83.5 | 82.7 | 87.9 |
| Transcript | 26 | 4.23 | 0.144 | 1.63 | 56.3 | 55.1 | 48.2 | 54.5 | 59.8 |
| Instance† | 26 | 4.23 | 0.144 | 1.63 | 85.3 | 84.6 | 79.9 | 79.8 | 86.1 |
| Instance | 100 | 4.45 | 0.139 | 1.63 | **89.3** | **88.4** | **83.9** | **84.2** | **89.5** |
| $\Delta_{\text{Instance−Ordered-class}}$ | | +0.71 | +0.003 | +0.14 | +1.2 | +1.4 | +0.4 | +1.5 | +1.6 |
| $\Delta_{\text{Instance−Transcript}}$ | | +0.22 | -0.005 | +0.14 | +33.0 | +33.3 | +35.7 | +29.7 | +29.7 |

Table 1: Comparative analysis of matching strategies on 50Salads. (#Q: number of queries.)

| Query | FLOP (G) | Time (s) | Para (M) | F1 @{10, 25, 50} | | | Edit | Acc. |
|---|---|---|---|---|---|---|---|---|
| ASFormer | 6.66 | 0.359 | 1.13 | 85.1 | 83.4 | 76.0 | 79.6 | 85.6 |
| Class Token | 4.54 | 0.139 | 1.63 | 85.7 | 84.5 | 78.9 | 79.7 | 86.3 |
| Average Pooling | 4.54 | 0.139 | 1.63 | **89.3** | **88.4** | **83.9** | **84.2** | **89.5** |
| $\Delta_{\text{(Average Pooling −Class Token)}}$ | 0 | 0 | 0 | +3.6 | +3.9 | +5.0 | +4.5 | +3.2 |

Table 2: Performance of different global queries on 50Salads.

ones, while Breakfast encompasses 1712 videos, which is the largest dataset. We conduct 4-fold cross-validation for GTEA and Breakfast and 5-fold cross-validation for 50Salads, consistent with previous research [21, 30, 43, 45, 15, 3, 42]. We assess accuracy for all frames as our key evaluation metric and report the segmental Edit score, and F1 scores for various Intersection over Union (IoU) thresholds (10%, 25%, 50%) denoted as F1@{10, 25, 50}.

**Implementation Details.** For the frame encoder, we use the pre-trained I3D [5] with fixed parameters, to obtain the frame-wise features with a dimension of 2048. The frame decoder allows any architecture designed for dense prediction tasks. In our paper, we utilize the ASFormer encoder [45], replicating its configuration with 11 layers and an output dimension $C$ of 64. For the Transformer decoder, the initialization starts with 90 queries for GTEA and 100 queries for 50Salads and Breakfast datasets, subsequently employing 10-layer Transformer decoders. Each decoder layer comprises three attention heads and a hidden dimension of 128. In the output heads, for mask and boundary prediction, we employ MLP layers with a hidden dimension of 64. As for Hungarian matching, $\lambda_{\text{focal}} = 5.0$ and $\lambda_{\text{dice}} = 1.0$. During training, we adopt Adam optimizer [23] with the step learning rate schedule in [19]. Initial learning rates are set to $5 \times 10^{-4}$ for GTEA and 50salads datasets, while $1 \times 10^{-3}$ for Breakfast dataset, incorporating a decay factor of 0.5. The training is with 300 epochs utilizing a batch size of 1. All experiments are conducted on an NVIDIA RTX 3090.

## 4.2 Ablation Studies and Analysis

We use **boldface** and underline for the best and second best performing methods in tables and indicate the performance improvements with $\Delta$. All ablation studies are conducted on the 50Salads dataset. More analysis is provided in the supplemental material.

**Analysis of matching strategies.** We evaluate the impacts of different matching strategies as summarized in Table 1. We explored three matching approaches: Ordered Class Matching, Transcript Matching, Instance Matching. Instance† utilizes the minimum feasible query count (26) to align computational costs with the other methods. In the ordered class matching model, deterministic query-to-class assignments provide a transparent optimization trajectory during training. Its commendable performance reflects its effectiveness. Instance†, which employs bipartite matching to correlate queries with unique segment instances, displays a slight decrease in performance relative to ordered class matching, with accuracy, F1 @10, and Edit distance decreased by 1.8%, 2.8%, and 2.9%, respectively. This modest divergence implies that fixed query-to-class mappings may benefit training. However, unlike static class numbers, increasing the number of queries from 26 to 100 enhances the performance of the instance-based method. In contrast, the transcript matching strategy underperforms, evidenced by its considerable performance metric deficits. This technique, which encodes only the sequential information into each query, predisposes the queries towards positional grouping rather than convergent model learning, indicating that this is insufficient for coalescing actions into meaningful query clusters.

**Effectiveness analysis of different kinds of global query.** The global query generates boundaries that offer class-agnostic segment proposals essential for the query-based voting mechanism. We explore two approaches for deriving the global query: a class token and an average pooling token. The class token, like a class token in ViT [14], is learned at the beginning of the Transformer decoder, whereas the average pooling token is aggregated from instance queries by an average pooling operation. The comparative results of these approaches are presented in Table 2. When evaluating different strategies, it demonstrates that the class token is inferior to the average pooling token. We speculate that the superior performance of the average pooling token can be attributed to its benefit from intermediate, query-based supervision, unlike the class token, which relies exclusively on

| Voting strategy | Time(s) | F1@{10,25,50} | | | Edit | Acc. |
|---|---|---|---|---|---|---|
| Frame-based(FV) | 0.187 | 85.3 | 84.8 | 77.2 | 78.6 | 87.7 |
| Query-based(QV) | 0.139 | **89.3** | **88.4** | **83.9** | **84.2** | **89.5** |
| $\Delta_{QV-FV}$ | -0.043 | +4.0 | +3.6 | +6.7 | +5.6 | +1.8 |

Table 3: Performance and efficiency of different voting strategies on 50Salads.

Figure 6: Query predictions and frame-wise results on 50Salads.

| Method | Time(s) | F1@{10,25,50} | | | Edit | Acc. |
|---|---|---|---|---|---|---|
| NMS | 0.138 | 89.1 | 88.4 | **84.0** | 83.8 | 89.1 |
| peak | 0.139 | **89.3** | 88.4 | 83.9 | **84.2** | **89.5** |
| $\Delta_{peak-NMS}$ | +0.001 | +0.2 | 0 | -0.1 | +0.4 | +0.4 |

Table 4: Different strategies on boundary generation on 50Salads.

| Boundary | F1@{10,25,50} | | | Edit | Acc. |
|---|---|---|---|---|---|
| Predict | 89.3 | 88.4 | 83.9 | 84.2 | 89.5 |
| GT | 91.8 | 91.8 | 90.2 | 88.3 | 95.9 |
| $\Delta_{GT-Predict}$ | +2.5 | +3.4 | +6.3 | +4.1 | +6.4 |

Table 5: Performance with predicted or ground-truth boundaries on 50Salads.

boundary supervision. This intermediate level of supervision likely results in finer temporal resolution and a more integrated global context, significantly improving the model's predictive accuracy.

**Why do we use query-based voting?** BaFormer efficiently compresses videos into sparse representations, thereby reducing computational demands. To derive the final frame-wise results from the predictions of query classes, masks, and boundaries, we employ two voting strategies: frame-based, and query-based voting. These strategies correspond to voting on boundaries by queries, and frames, respectively (details are provided in supplemental material). The results of these voting strategies are presented in Table 3. Specifically, the query-based voting mechanism identifies the query contributing most significantly to a segment, treating each query as an integral unit. Conversely, the frame-based voting method, detailed in the supplemental material, determines the segment's class label based on the majority class across frames within the segment. According to the comparison detailed in Table 3, query-based voting significantly surpasses frame-based voting, showcasing improvements in running time and evaluation metrics. Specifically, accuracy obtains an enhancement of 1.8%. The edit score is increased by 5.6%, and F1@50 is improved by 6.7%.

Further, we identify the reasons why query-based voting enhances the results and why it is superior to frame-based voting in Figure 6. Figure 6 comprises two parts. Specifically, to generate the top part, we binarize the predicted masks from the final transformer decoder layer $\mathbf{P}_L^m \in \mathbb{R}^{M \times T}$ to 0 or 1 with a threshold 0.5. Thus, the positions with a value of 1 indicate the presence of actions corresponding to the query class. Next, we apply different colors to these positions and each color represents a single action. Then query masks are stacked to form the top part. In the bottom part, we present video segment results of frame-based and query-based voting, along with the ground truth. The red arrow points towards a specific query within a segment proposal, illustrated between two vertical black dashed lines. The red dashed box shows the segment results within the proposal. Query-based voting can identify new action segments, in contrast to frame-based voting, which is limited to recognizing only the predefined action classes within a segment. As demonstrated in Figure 6, we highlight intervals where new actions are discerned that would not be apparent through smoothing of frame-wise results. We hypothesize that this is due to the voting process among segmental candidates, which is independent of class probability and focuses instead on the number of frames. This approach can rectify misclassifications inherent in frame-wise analysis that overemphasizes class probability.

**Different strategies for boundary generation.** To obtain boundaries quickly, we experiment with two techniques: Non-Maximum Suppression (NMS) and peak choice, with the corresponding results presented in Table 4. Peak choice focuses on identifying boundaries within a local context. In contrast, NMS initially identifies boundaries with the maximum probability on a global scale and subsequently eliminates other boundary candidates at a local level. The analysis reveals that they yield similar outcomes and running time.

**How well would our approach perform if we had *perfect* boundaries?** We also conduct an analysis related to class-agnostic boundaries. Given that boundary quality is crucial to our process, we aim to quantify the potential performance improvement if "perfect" boundaries are available. To this end,

| S | Method | Time (s) | FLOPs (G) | Param (M) | GTEA | | | | | 50Salads | | | | | Breakfast | | | | |
|---|---|---|---|---|---|---|---|---|---|---|---|---|---|---|---|---|---|---|---|
| | | | | | F1@{10,25,50} | | | Edit | Acc. | F1@{10,25,50} | | | Edit | Acc. | F1@{10,25,50} | | | Edit | Acc. |
| Multiple | MSTCN [15] 2019 | 0.094 | 4.59 | 0.80 | 85.8 | 83.4 | 69.8 | 79.0 | 76.3 | 76.3 | 74.0 | 64.5 | 67.9 | 80.7 | 52.6 | 48.1 | 37.9 | 61.7 | 66.3 |
| | SSTDA [7] 2020 | 0.173 | 9.37 | 0.80 | 90.0 | 89.1 | 78.0 | 86.2 | 79.8 | 83.0 | 81.5 | 73.8 | 75.8 | 83.2 | 75.0 | 69.1 | 55.2 | 73.7 | 70.2 |
| | BCN [43] 2020 | 0.152 | 73.54 | 12.77 | 88.5 | 87.1 | 77.3 | 84.4 | 79.8 | 82.3 | 81.3 | 74.0 | 74.3 | 84.4 | 68.7 | 65.5 | 55.0 | 66.2 | 70.4 |
| | HASR [1] 2021 | 0.217 | 29.02 | 19.17 | 90.9 | 88.6 | 76.4 | 87.5 | 78.7 | 86.6 | 85.7 | 78.5 | 81.0 | 83.9 | 74.7 | 69.5 | 57.0 | 71.9 | 69.4 |
| | DTGRM [41] 2021 | 0.261 | 3.75 | 0.73 | 87.8 | 86.6 | 72.9 | 83.0 | 77.6 | 79.1 | 75.9 | 66.1 | 72.0 | 80.0 | 68.7 | 61.9 | 46.6 | 68.9 | 68.3 |
| | ASRF [21] 2021 | 0.163 | 7.43 | 1.30 | 89.4 | 87.8 | 79.8 | 83.7 | 77.3 | 84.9 | 83.5 | 77.3 | 79.3 | 84.5 | 74.3 | 68.9 | 56.1 | 72.4 | 67.6 |
| | Gao *et al* [17] 2021 | - | - | - | 89.9 | 87.3 | 75.8 | 84.6 | 78.5 | 80.3 | 78.0 | 69.8 | 73.4 | 82.2 | 74.9 | 69.0 | 55.2 | 73.3 | 70.7 |
| | ASFormer [45] 2021 | 0.359 | 6.66 | 1.13 | 90.1 | 88.8 | 79.2 | 84.6 | 79.7 | 85.1 | 83.4 | 76.0 | 79.6 | 85.6 | 76.0 | 70.6 | 57.4 | 75.0 | 73.5 |
| | UARL [6] 2022 | - | - | - | 92.7 | 91.5 | 82.8 | 88.1 | 79.6 | 85.3 | 83.5 | 77.8 | 78.2 | 84.1 | 65.2 | 59.4 | 47.4 | 66.2 | 67.8 |
| | DTL [44] 2022 | 0.403 | 6.66 | 1.13 | - | - | - | - | - | 87.1 | 85.7 | 78.5 | 80.5 | 86.9 | 78.8 | 74.5 | 62.9 | 77.7 | 75.8 |
| | RTK [22] 2023 | - | - | - | 91.2 | 90.6 | 83.4 | 87.9 | 80.3 | 87.4 | 86.1 | 79.5 | 81.4 | 85.9 | 76.9 | 72.4 | 60.5 | 76.1 | 73.3 |
| | LtContext [2] 2023 | 0.202 | 8.31 | 0.66 | - | - | - | - | - | 89.4 | 87.7 | 82.0 | 83.2 | 87.7 | 77.6 | 72.6 | 60.1 | 77.0 | 74.2 |
| | DiffAct [32] 2023 | 2.306 | 43.94 | 1.21 | 92.5 | 91.5 | 84.7 | 89.6 | 82.2 | 90.1 | 89.2 | 83.7 | 85.0 | 88.9 | 80.3 | 75.9 | 64.6 | 78.4 | 76.4 |
| | KARI [18] 2023 | - | - | - | - | - | - | - | - | 85.4 | 83.8 | 77.4 | 79.9 | 85.3 | 78.8 | 73.7 | 60.8 | 77.8 | 74.0 |
| Single | UVAST† [3] 2022 | 0.577 | 3.86 | 1.27 | 77.1 | 69.7 | 54.2 | 90.5 | 62.2 | 86.2 | 81.2 | 70.4 | 83.9 | 79.5 | 76.7 | 70.0 | 56.6 | 77.2 | 68.2 |
| | UVAST [3] 2022 | 480.888 | 3.06 | 1.10 | 92.7 | 91.3 | 81.0 | 92.1 | 80.2 | 89.1 | 87.6 | 81.7 | 83.9 | 87.4 | 76.9 | 71.5 | 58.0 | 77.1 | 69.7 |
| | UVAST‡ [3] 2022 | 1.765 | 3.86 | 1.27 | 82.9 | 79.4 | 64.7 | 90.5 | 69.8 | 88.9 | 87.0 | 78.5 | 83.9 | 84.5 | 76.9 | 71.5 | 58.0 | 77.1 | 69.7 |
| | BaFormer (ours) | 0.139 | 4.54 | 1.63 | 92.0 | 91.3 | 83.5 | 88.7 | 83.0 | 89.3 | 88.4 | 83.9 | 84.2 | 89.5 | 79.2 | 74.9 | 63.2 | 77.3 | 76.6 |

Table 6: Performance on GTEA, 50Salads, and Breakfast datasets. In terms of running time, BaFormer outperforms all methods except MSTCN. As for accuracy, BaFormer achieves comparable or better results. UVAST†, UVAST, and UVAST‡ represent UVAST with alignment decoder, Viterbi, and FIFA. All FLOPs and running time are evaluated on 50Salads using the official codes in a consistent environment. We omit the running time and FLOPs on GTEA and Breakfast for simplicity as they are proportional to video length.

| Method | Base | FLOPs(G) | Time(s) | Parameters(M) | F1@{10, 25, 50} | | | Edit | Acc. |
|---|---|---|---|---|---|---|---|---|---|
| SSTCN | | 1.71 | 0.035 | 0.30 | 27.0 | 25.3 | 21.5 | 20.5 | 78.2 |
| MSTCN | CNN | 4.59 | 0.094 | 0.80 | 76.3 | 74.0 | 64.5 | 67.9 | 80.7 |
| BaFormer | | 4.02 | 0.074 | 1.55 | **84.3** | **83.1** | **75.8** | **78.4** | **84.5** |
| $\Delta_{BaFormer-SSTCN}$ | - | +2.31 | +0.039 | +1.25 | +57.3 | +57.8 | +54.3 | +57.9 | + 6.3 |
| ASFormer(Encoder) | | 2.23 | 0.110 | 0.38 | 53.1 | 51.4 | 47.0 | 43.3 | 85.7 |
| DiffAct(1 step Decoder) | Transformer | 7.78 | 1.647 | 1.21 | 48.3 | 45.3 | 35.6 | 36.5 | 74.2 |
| UVAST(Alignment) | | 3.86 | 0.577 | 1.27 | 86.2 | 81.2 | 70.4 | 83.9 | 79.5 |
| BaFormer | | 4.54 | 0.139 | 1.63 | **89.3** | **88.4** | **83.9** | **84.2** | **89.5** |
| $\Delta_{BaFormer-ASFormer(Encoder)}$ | | +2.31 | +0.029 | +1.25 | +36.2 | +37.0 | +36.9 | +40.9 | + 3.8 |

Table 7: Performance of methods with similar running time, employing the CNN or Transformer based frame decoder on the 50Salads dataset. To achieve comparable running time, DiffAct (1 step) is adapted with an encoder and a single-step decoder, and ASFormer with an encoder only is included.

we run BaFormer using ground truth boundaries on the 50Salads dataset instead of our predicted class-agnostic boundaries, as shown in Table 5. All the metrics have improved significantly. This indicates that our approach yields more promising results by higher-quality class-agnostic boundaries, which can be one of the future directions.

### 4.3 Comparisons with the State of the Arts

In Table 6, we categorize methods by multi-stage and single-stage. Our main comparison is with the single-stage methods since our BaFormer adopts the same single-stage frame-wise decoder. Compared with UVAST with different alignment modes, BaFormer demonstrates superior efficiency, achieving the shortest processing time while outperforming it in most metrics. Notably, BaFormer requires only 24% of the time compared to UVAST with an alignment decoder (UVAST†), while enhances accuracy by 20.8%, 10%, and 8.4% on the GTEA, 50Salads, and Breakfast datasets, respectively. These results underscore the efficacy of BaFormer in decoupling frame-wise prediction into query masks and class prediction. Overall, BaFormer surpasses all multi-stage methods in terms of accuracy and achieves comparable results in F1 and Edit scores on all datasets. As for efficiency, compared to state-of-the-art multi-stage methods, *i.e.*, DiffAct, our method operates with only 6% of inference time and 10% of FLOPs, demonstrating a highly efficient approach to TAS.

**Comparison with the methods with similar Running time.** We compare our method with several others that exhibit similar running time in Table 7, focusing on both CNN and Transformer-based frame-wise decoder. Specifically, 'CNN based frame-wise deocder' refers to the single-stage MSTCN structure, and 'Transformer based frame-wise decoder' refers to the single stage of ASFormer structure. Within the CNN based frame-wise decoder, *i.e.*, SSTCN [15], BaFormer achieves outstanding

enhancements, surpassing MSTCN by 3.8% in accuracy, 10.5% in Edit score, and 8.0% in F1@50. Notably, BaFormer significantly augments SSTCN's performance, yielding a 54.3% increase in F1@50. This comparison underscores BaFormer's single-stage capability to achieve high performance efficiently, maintaining the same inference time. When leveraging a Transformer as the frame-wise decoder, we modify the number of stages in multi-stage methods to match the inference time of BaFormer. This adjustment leads to a new version of ASFormer with a single encoder and DiffAct with a simplified step decoder. Among these methods, BaFormer emerges as the top performer. This signifies that achieving frame-wise results through the decoupling of query masks and class prediction is more effective than direct frame-wise prediction.

## 5   Limitations and Conclusion

BaFormer takes the query-based Transformer framework, which is known for its slow training convergence and data thirsty. We observe that training query-based Transformer converges slower than training frame-wise approaches. For the discontinuous binary mask predictions, it may also be attributed to the limited data of action segmentation benchmarks. We hope that these factors can inspire future work.

In conclusion, we introduce BaFormer, a novel boundary-aware, query-based approach for efficient temporal action segmentation. Contrasting most of the previous methods that rely on multi-stage or multi-step processing, BaFormer employs a one-step strategy. It simultaneously predicts the query-wise class and mask, while yielding global boundary prediction for segment proposals. We apply query-based voting for segment proposal classification. BaFormer offers a unique perspective for addressing TAS challenges by integrating grouping and classification techniques, representing a novel perception in the temporal action segmentation paradigm, emphasizing both efficiency and accuracy.

## Acknowledgments

We thank all reviewers for valuable comments and suggestions. The work was supported in part by US National Science Foundation Grants 2006665, 2128350, and 2331769. This work is also supported in part by Air Force Research Laboratory FA 9550-23-2-0002. The support of these agencies is gratefully acknowledged. Any opinions, findings, and conclusions, or recommendations expressed in this material are those of the authors and do not necessarily reflect the views of the National Science Foundation or the United States Air Force.

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

# Appendix

## A    Broader impact

We assert that our proposed method processes temporal action segmentation efficiently while strictly preserving privacy by utilizing features rather than raw video. Our work not only opens new horizons for the academic community but also advances the task towards real-world application. We also acknowledge that ethical concerns may be caused by the unsatisfactory boundary detection. Nonetheless, safety and reliability will be our top priorities when we deploy this system in real-world applications.

## B    Notations

To clarify, we denote $l$ as the index for the transformer decoder layer and $i$ as the index for the query. Subscripts indicate the indices of parameters, while superscripts serve as abbreviations representing parameter types. For instance, the superscript $m$ in $\mathbf{P}^m$ stands for '**m**ask.'

| Notation | Meaning |
|---|---|
| $\mathbf{F}^e$ | Video features extracted from frame encoder |
| $\mathbf{F}^d$ | Outputs of frame-wise encoder-decoder module |
| $\mathbf{F}$ | Features collections from $L$ frame decoder layers |
| $\mathbf{f}_l$ | Output features of $l^{\text{th}}$ frame decoder layer |
| $\mathbf{Q}_0$ | $M$ trainable instance queries with random initialization |
| $\mathbf{Q}_l$ | Instance query embedding from the $l^{\text{th}}$ layer |
| $\mathbf{P}_l^c$ | Query class prediction from the $l^{\text{th}}$ layer (the $c$ represents '**c**lass') |
| $\mathbf{P}_l^m$ | Query masks prediction from the $l^{\text{th}}$ layer (the $m$ represents '**m**ask') |
| $\mathbf{P}_l^b$ | Boundary predictions from the $l^{\text{th}}$ layer (the $b$ represents '**b**oundary') |
| $\mathbf{p}_i^c$ | The $i^{\text{th}}$ query class prediction in $\mathbf{P}^c$ (the index $l$ is committed in $\mathbf{P}^c$ for simplicity) |
| $\mathbf{p}_i^m$ | The $i^{\text{th}}$ query mask prediction in $\mathbf{P}^m$ (the index $l$ is committed in $\mathbf{P}^m$ for simplicity) |
| $\mathrm{p}_t^b$ | The boundary prediction in the $t^{\text{th}}$ frame |
| $\varphi^c$ | The head for predicting query classes |
| $\varphi^m$ | The head for predicting query masks |
| $\varphi^b$ | The head for predicting boundaries |
| $y_i^c$ | The ground-truth action class for the $i^{\text{th}}$ query |
| $\mathbf{y}_i^m$ | The ground-truth binary mask for the $i^{\text{th}}$ query |
| $\mathbf{y}^b$ | The ground-truth boundaries |

## C    Other implementation details

### C.1    Loss for ordered class and transcript matching

Similar to instance matching, we model ground-truth boundaries as $\mathbf{y}^b \in \mathbb{R}^T$, a heatmap generated by applying a Gaussian distribution to the binary boundary mask. For ground truths of query classes and masks, we employ different strategies: In ordered class matching, segments are derived according to categorical order. In contrast, transcript matching segments are determined by the action sequence outlined in the video transcript. Consequently, we represent ground-truth segments as class-mask pairs $\{(y_i^c, \mathbf{y}_i^m)\}_{i=1}^N$, with $y_i^c \in \mathbb{R}$ denoting the action class, $\mathbf{y}_i^m \in \mathbb{R}^T$ as the corresponding binary mask, and $N$ as the total number of segments in the video. In ordered class matching, the $N = K$, where $K$ is the total number of action classes in the dataset. In the transcript matching, the $N$ is the number of actions in the video transcript.

The predicted outputs for classes and masks, denoted as $\mathbf{P}^c$ and $\mathbf{P}^m$, are organized into class-mask pairs $\{(\mathbf{p}_i^c, \mathbf{p}_i^m)\}_{i=1}^{M}$, where $\mathbf{p}_i^c \in \mathbb{R}^{K+1}$ and $\mathbf{p}_i^m \in \mathbb{R}^T$ represent the $i^{\text{th}}$ query's predicted class probabilities (including a no-label class) and mask probabilities, respectively. We define $\mathbf{p}_{(i,j)}^c$ as the probability value in the $i^{\text{th}}$ query class prediction $\mathbf{p}_i^c$ indexed by the $j^{\text{th}}$ class. We define the cost when matching a pair of mask $\mathbf{a} \in \mathbb{R}^T$ and $\mathbf{b} \in \mathbb{R}^T$ as:

$$\mathcal{L}_{\text{mask}}(\mathbf{a}, \mathbf{b}) = \lambda_{\text{focal}}\mathcal{L}_{\text{focal}}(\mathbf{a}, \mathbf{b}) + \lambda_{\text{dice}}\mathcal{L}_{\text{dice}}(\mathbf{a}, \mathbf{b}), \tag{8}$$

where the $\lambda_{\text{focal}}$ and $\lambda_{\text{dice}}$ is the loss weight of focal loss [31] and dice loss [35].

As a result, the loss function in ordered class and transcript matching can be formulated as follows:

$$\mathcal{L} = \sum_{i=1}^{N} \left( \mathcal{L}_{\text{mask}}(\mathbf{p}_i^m, \mathbf{y}_i^m) - \log \mathbf{p}_{(i,y_i^c)}^c \right) - \sum_{t=1}^{T} y_t^b \log p_t^b, \tag{9}$$

where $M$ is the number of queries, $T$ is the length of the video, $y_t^b$ is the $t^{\text{th}}$ element of $\mathbf{y}^b$, and $p_t^b$ is the $t^{\text{th}}$ element of $\mathbf{P}^b$, which is the prediction of boundary probability.

## C.2 Boundary-aware frame-wise voting

Different from the query-based voting, we also explore frame-wise voting, a more direct approach when starting with frame-wise results. The details of the algorithm are in Algorithm 2.

---

**Algorithm 2:** Boundary-aware Frame Voting

---

**Input:** Probability of query class $\mathbf{P}^c \in \mathbb{R}^{M \times (K+1)}$; Probability of query masks $\mathbf{P}^m \in \mathbb{R}^{M \times T}$;
Boundary probability: $\mathbf{P}^b = \{p_t^b\}_{t=1}^{T}$, where $p_t^b \in \mathbb{R}$ is the boundary probability in the $t^{\text{th}}$ frame.
**Output:** Frame-wise segmentation: $\mathbf{S} \in \mathbb{R}^T$.

1  Initialize $\mathbf{S} \in \mathbb{R}^T$ with all zeros
2  $\mathbf{P}^s = (\mathbf{P}^m)^\intercal (\mathbf{P}^c[:, :K]) \in \mathbb{R}^{T \times K}$
3  $\mathbf{S}_0 = \text{argmax}(\mathbf{P}^s) \in \mathbb{R}^T$
4  $\mathbf{B} = \{b_i\}_{i=1}^{N_b} \leftarrow \text{sort}(\{1, T\} \bigcup \{t | (p_t^b > p_{t-1}^b) \& (p_t^b > p_{t+1}^b), 1 < t < T\})$
5  **for** $i = 1, 2, ..., N_b - 1$ **do**
6     $\text{act\_id} = \text{FindMajorityElement}(\mathbf{S}_0[b_i : b_{i+1}])$
7     $\mathbf{S}[b_i : b_{i+1}] = \text{act\_id}$
8  **end**

---

# D  Impact of parameters

## D.1  Number of transformer decoder layers

Table 8 explores the impact of varying the number of transformer decoder layers on the 50Salads dataset and reveals a trend correlating the number of layers with performance metrics. Reducing the transformer decoder layers to 3 results in a significant reduction of FLOPs to 2.97G and a minimal change in running time, but it leads to decreased performance across all metrics, indicating a trade-off between a more lightweight model and diminished robustness.

A further increase to 10 layers results in the best performance metrics, surpassing the model with 8 layers by 1.1% on accuracy, indicating that the additional depth provides a beneficial complexity that enhances the model's learning capability. Overall, the results suggests that while increasing the number of transformer decoder layers incurs more computational cost, it also significantly boosts performance, indicating a trade-off between efficiency and efficacy.

## D.2  Query quantity

Table 9 provides an insightful look into how the quantity of queries impacts both the computational cost and the performance of a model. Starting with 50 queries, we see a computational cost of

| # Transformer decoder layer | FLOPs (G) | Running time(s) | #Parameter (M) | F1 @{10, 25, 50} | | | Edit | Acc. |
|---|---|---|---|---|---|---|---|---|
| 3 | 2.97 | 0.133 | 0.78 | 82.8 | 81.6 | 76.5 | 77.2 | 84.2 |
| 5 | 3.42 | 0.136 | 1.02 | 86.4 | 85.6 | 80.7 | 79.8 | 86.3 |
| 8 | 4.09 | 0.137 | 1.38 | 88.6 | 87.1 | 83.1 | 83.4 | 88.4 |
| 10 | 4.45 | 0.139 | 1.63 | **89.3** | **88.4** | **83.9** | **84.2** | **89.5** |

Table 8: Results of different numbers of Transformer decoder layers on 50Salads.

| # query | FLOPs (G) | Running time(s) | # Parameter (M) | F1 @{10, 25, 50} | | | Edit | Acc. |
|---|---|---|---|---|---|---|---|---|
| 50 | 4.26 | 0.138 | 1.629 | 87.8 | 87.2 | 82.8 | 82.8 | 87.1 |
| 70 | 4.28 | 0.139 | 1.630 | 88.4 | 87.5 | 83.4 | 82.9 | 87.9 |
| 100 | 4.45 | 0.139 | 1.632 | **89.3** | **88.4** | **83.9** | **84.2** | **89.5** |
| 120 | 4.56 | 0.139 | 1.633 | 86.6 | 85.7 | 81.1 | 81.3 | 87.0 |
| 150 | 4.60 | 0.139 | 1.636 | 84.3 | 82.8 | 77.2 | 78.4 | 84.4 |

Table 9: Influence of query quantity on 50Salads.

4.26G FLOPs and an accuracy of 87.1%. Increasing the query count to 70 results in a marginal increase in FLOPs to 4.28G, but a notable improvement in accuracy to 87.9%. This indicates that a slight increase in the number of queries can enhance the model's ability without significantly raising computational costs. However, the model's performance peaks at 100 queries, achieving the highest accuracy of 89.5% along with the best F1 and Edit scores, at a computational cost of 4.45G FLOPs. This suggests that there is an optimal query range where the model's performance is maximized. Interestingly, beyond this point, as the number of queries continues to rise to 120 and 150, there is a diminishing return in terms of accuracy, showing that additional queries beyond a certain threshold may instead lead to overfitting.

# E    Structure and loss

## E.1    Single/Multiple features

**Different connection between frame decoder and transformer decoder.** We investigate diverse strategies for integrating the frame decoder with the transformer decoder modules. In the simplest form, a uniform connectivity strategy is employed, wherein the transformer decoder ingests shared video features originating from a singular level of the frame decoder, as illustrated in Figure 7(a). Conversely, a more sophisticated approach is explored, wherein the transformer decoder harnesses a hierarchy of video features, sourced from multiple layers within the frame decoder, to enrich the representational capacity of the system, as demonstrated in Figure 7(b).

The results of these connectivity strategies are detailed in Table 10. We include the ASFormer as a baseline for comparison. Regardless of whether auxiliary losses are applied or not, the adoption of multi-level features yields substantial and noteworthy performance enhancements. These performance improvements come at a minimal cost in terms of additional parameters and FLOPs. Specifically, we observe an increase in accuracy by $5.5\%$ and F1@10 by $6.0\%$, along with a corresponding gain in accuracy by $3.9\%$ and F1@10 by $3.9\%$ in the presence and absence of auxiliary losses, respectively. This phenomenon can be attributed to the mutually beneficial relationship between query and mask predictions. When the transformer decoder and frame decoder operate independently, there is a deficiency in the exchange of crucial information, impeding the training of query-based mask classification methods.

## E.2    Auxiliary loss

To rigorously analyze the impact of various loss functions on our method, we provide comparative results in Table 10, detailing performance metrics both with and without auxiliary losses. These

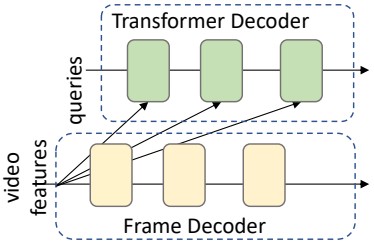
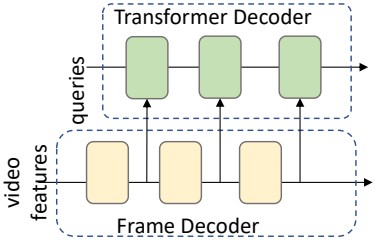

(a) Single-level features connection      (b) Multi-level features connection

Figure 7: (a) and (b) illustrate the single-level and multi-level feature connection strategies, respectively. In (a), a single-level feature from the frame decoder is shared with the transformer decoder layers. While (b) involves the integration of multi-level features from various layers of the frame decoder. (Note: Mask inputs have been omitted for simplicity.)

| | Feature | Auxiliary loss | #Parameter (M) | Running Time(s) | FLOPs (G) | F1 @{10, 25,50} | | | Edit | Acc. |
|---|---|---|---|---|---|---|---|---|---|---|
| ASFormer | - | - | 1.13 | 0.359 | 6.66 | 85.1 | 83.4 | 76.0 | 79.6 | 85.6 |
| BaFormer | single | no | 1.51 | 0.137 | 3.83 | 82.6 | 81.8 | 75.6 | 76.6 | 83.5 |
| | | yes | | | | 83.3 | 82.2 | 76.5 | 78.6 | 84.0 |
| | multiple | no | 1.63 | 0.139 | 4.45 | 86.5 | 85.3 | 81.2 | 80.4 | 87.4 |
| | | yes | | | | **89.3** | **88.4** | **83.9** | **84.2** | **89.5** |

Table 10: Comparative analysis of the effect of feature connections, *i.e.*, single or multiple, on 50Salads and the use of auxiliary loss.

results indicate that auxiliary losses enhance performance in both single-level and multi-level feature representations. Notably, applying auxiliary losses in multi-level features yields a substantial improvement in accuracy (2.1%), surpassing the 0.5% accuracy gain observed in single-level features. This suggests that auxiliary losses have a more pronounced effect on multi-level feature representations.

## F   More detailed efficiency comparison and experiments

### F.1   Methods with various stages.

Table 11 presents a comparative analysis of various multi-stage/step methods in contrast to BaFormer for Temporal Action Segmentation (TAS). The methods compared include MSTCN with varying stages, ASFormer with different numbers of decoders, and DiffAct with varying inference steps. We have calculated the FLOPs and parameter count utilizing the official code repositories. We employed the DiffuAct model, as it is reported to achieve the final results in their publication, to determine the FLOPs and parameter metrics.

BaFormer has moderate FLOPs (4.45 G) and parameter count (1.64M), which stands between the lightest and heaviest models compared. It is considerably more efficient than the most computationally intensive models like DiffAct with 25 inference steps. MSTCN and ASFormer models show a trend where increased stages or decoders generally lead to better performance, but also increased computational costs. DiffAct's performance improves significantly with the number of inference steps, yet this comes at the cost of a substantial increase in FLOPs.

In conclusion, BaFormer achieves a balance between computational efficiency and performance, making it a compelling choice for applications where both are critical considerations and presents a competitive alternative to existing multi-stage/step methods, offering a reduction in computational demands while maintaining high accuracy and outperforming in critical metrics like F1 score and Edit distance. This suggests that BaFormer's approach to integrating hierarchical tokens and utilizing a boundary-aware transformer architecture is effective for TAS tasks.

| Method | #Stage /Step | FLOPs (G) | Running time(s) | #Parameter (M) | F1 @{10, 25, 50} | | | Edit | Acc. |
|---|---|---|---|---|---|---|---|---|---|
| MSTCN[1] | 1 | 1.71 | 0.080 | 0.30 | 27.0 | 25.3 | 21.5 | 20.5 | 78.2 |
| MSTCN[2] | 2 | 2.67 | 0.086 | 0.47 | 55.5 | 52.9 | 47.3 | 47.9 | 79.8 |
| MSTCN[3] | 3 | 3.63 | 0.092 | 0.63 | 71.5 | 68.6 | 61.1 | 64.0 | 78.6 |
| MSTCN[4] | 4 | 4.59 | 0.094 | 0.80 | 76.3 | 74.0 | 64.5 | 67.9 | 80.7 |
| ASFormer[1] | 1 | 2.23 | 0.168 | 0.38 | 53.1 | 51.4 | 47.0 | 43.3 | 85.7 |
| ASFormer[2] | 2 | 3.71 | 0.235 | 0.63 | 79.5 | 77.4 | 71.6 | 71.5 | 86.8 |
| ASFormer[3] | 3 | 5.19 | 0.299 | 0.88 | 83.9 | 82.8 | 76.8 | 76.7 | 86.8 |
| ASFormer[4] | 4 | 6.66 | 0.359 | 1.13 | 85.1 | 83.4 | 76.0 | 79.6 | 85.6 |
| DiffAct[1] | 1 | 7.78 | 1.647 | 1.21 | 64.9 | 63.8 | 59.3 | 56.5 | 88.6 |
| DiffAct[4] | 4 | 12.30 | 1.784 | 1.21 | 87.6 | 86.6 | 81.2 | 82.1 | 89.1 |
| DiffAct[8] | 8 | 18.33 | 1.887 | 1.21 | 89.3 | 88.3 | 83.1 | 83.5 | 89.0 |
| DiffAct[16] | 16 | 30.38 | 2.043 | 1.21 | 90.0 | 88.8 | 83.3 | 84.5 | 89.0 |
| DiffAct[25] | 25 | 43.94 | 2.306 | 1.21 | 90.1 | 89.2 | 83.7 | 85.0 | 88.9 |
| BaFormer | 1 | 4.45 | 0.139 | 1.63 | 89.3 | 88.4 | 83.9 | 84.2 | 89.5 |

Table 11: Comparative overview of multi-stage/step methods versus BaFormer on 50Salads. Here, "MSTCN[n]" and "ASFormer[n]" denotes a model with $n$ processing stages, while "DiffAct[n]" signifies a model with "$n$" decoder steps.

| Segment length | 0∼1000 | 1001∼2000 | 2001∼3100 |
|---|---|---|---|
| Frame-based | 87.68 | 90.98 | 92.47 |
| Query-based | 86.85 | 92.01 | 95.38 |

Table 12: Accuracy for action segments of different lengths, comparing frame-based and query-based methods on the 50Salads dataset.

## F.2 Performance on segments with different lengths

To further compare the results of the frame-based and query-based voting methods, we propose analyzing the outcomes at the segment level rather than across entire videos. We conduct experiments on action segments of varying lengths based on the ground truth annotations of 50Salads dataset, and the results are summarized below in Table 12. We observe that frame-based methods perform slightly better on shorter segments, while query-based methods demonstrate higher accuracy on longer segments. We attribute this difference to the voting mechanism. Specifically, when applying the same boundaries for both methods, the query-based approach relies on query-level classification and masks. This strategy captures information at the segment level, as opposed to frame-based methods, which depend on individual frame-level predictions, leading to a distinction in performance.

## G More visualization

In the 50Salads dataset, we present enhanced visualizations for a more comprehensive analysis. Each subfigure's upper portion depicts the outcomes of instance segmentation, while the lower portion compares the segmentation results without boundary integration ("F"), the results achieved by BaFormer ("S"), and the ground truth ("GT"). The absence of boundary consideration ("F"), where the predicted instance token class probability is directly combined with the mask probability, leads to excessive over-segmentation and suboptimal boundary delineation. This effect is attributable to the frame-wise basis of mask prediction, which, when applied directly, perpetuates the over-segmentation issue. "F" with "S" illustrates that BaFormer's advancements are realized through the reduction of over-segmentation and the refinement of boundaries. This improvement stems from the more continuous nature of segment proposals informed by boundary predictions, enhancing the assignment of actions to each proposal. The transition from the results of instance segmentation to frame-wise

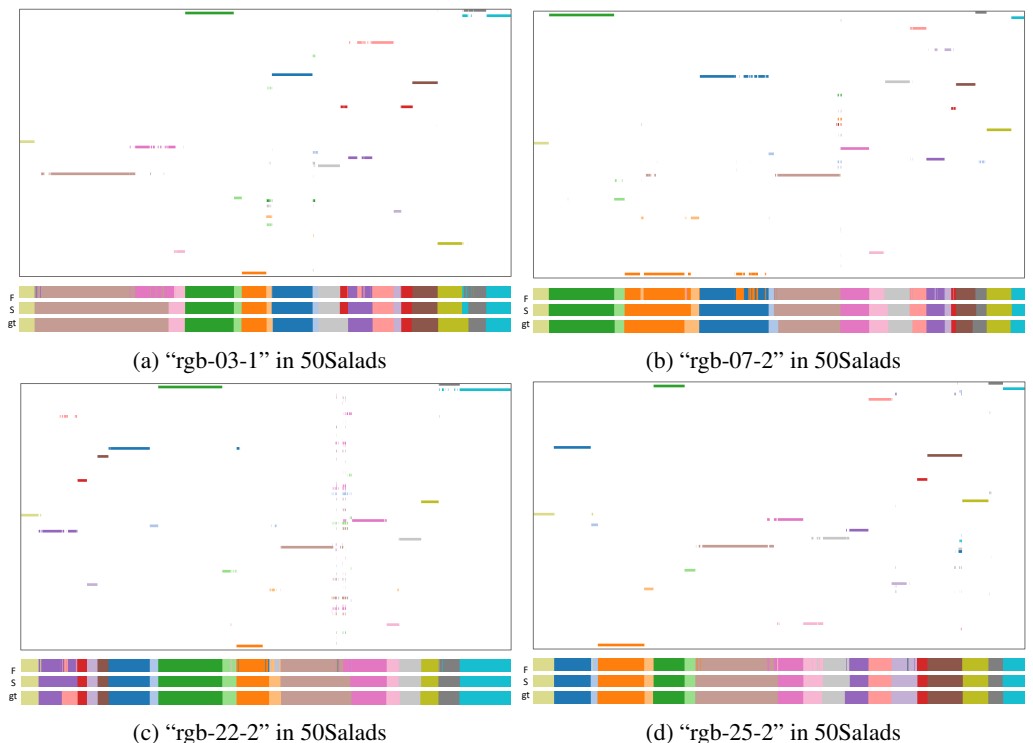

(a) "rgb-03-1" in 50Salads

(b) "rgb-07-2" in 50Salads

(c) "rgb-22-2" in 50Salads

(d) "rgb-25-2" in 50Salads

Figure 8: Visualization of the 50Salads dataset. Each subfigure presents a comparison of instance segmentation and frame-wise results. "F" indicates the absence of boundary utilization. "S" signifies its inclusion. "gt" represents the ground truth.

outcomes elucidates the efficacy of the majority voting mechanism. (Note: Boundary prediction is not depicted in these visualizations.)

