# OpenReview forum: "Efficient Temporal Action Segmentation via Boundary-aware Query Voting"
_NeurIPS.cc/2024/Conference — NeurIPS 2024 poster_

### Official Review · Reviewer_KTbs · 2024-07-10

**Soundness:** 3
**Presentation:** 3
**Contribution:** 2
**Rating:** 5
**Confidence:** 3

**Summary:**

The paper introduces BaFormer, a novel Transformer network designed to improve the efficiency of Temporal Action Segmentation (TAS) while maintaining high performance. BaFormer tokenizes video segments into instance tokens and utilizes instance queries for segmentation along with a global query for boundary prediction, resulting in continuous segment proposals. The model employs a voting strategy during inference to classify segments based on instance segmentation, significantly reducing computational costs. Experiments on popular TAS benchmarks demonstrate BaFormer's superior efficiency with comparable or better accuracy than state-of-the-art methods like DiffAct.

**Strengths:**

In general, the proposed method is somehow simple but effective, and I am satisfied with the paper. The proposed Beformer utilizes a simple structure but can achieve performance gain and inference speed gain over previous methods.

The proposed method also has detailed ablation studies, which can verify the effectiveness of the proposed BeFormer. Though BeFormer is a single-stage method, it performs comparable to the most recent two-stage methods.

**Weaknesses:**

However, there are some small concerns about the paper.

1. Despite the amazing performance, I think the novelty is limited. The author utilizes existing frameworks, including existing frame encoder and ASFormer encoder, also, the decoder is a common query-based decoder. The main novelty exists in the boundary-aware query voting part. However, it is a post-processing process. Thus in general the novelty is minor.

2. In Table 1 (teaser), the author claims that BaFormer with SSTCN and ASFormer backbones performs better than the original module with less running time. However, in Table 7, they have more FLOPs than the original methods but with less inference speed. Does the extra running time come from the NMS post-processing? If so, I think it is somehow unfair here. Maybe the author could list the modeling forward time and the post-processing time for a better and fair comparison. Otherwise, it is somehow confusing here.

**Questions:**

Please mainly see the weaknesses section for my problems.

**Limitations:**

the authors have adequately addressed the limitations

---

> ### Author Rebuttal · Authors · 2024-08-07
>
> We thank the reviewer for investing time in reviewing our work and acknowledging our contributions.
>
> **1. For Weakness 1**
>
> (**utilizes existing frameworks**) We understand the reviewer's concerns regarding our use of existing frameworks; however, we employ them solely to achieve the desired functionality. An important aspect we need to clarify is that we focus on how to adopt a new pipeline, i.e., query-based pipeline, into efficient TAS rather than some specific network design.  To achieve more efficient TAS, we experimentally find that query-based processing can reduce dense information, thus reducing FLOPs. Therefore, we adopt query-based prediction from a DETR-style model to improve the running time. This lets us establish a new paradigm in TAS, different from the frame prediction in previous work. Our main effort is how to adopt the DETR-style model into TAS to achieve the expected performance, rather than to design some specific network structures. Specifically, we focus on introducing functional modules with effective organization, i.e., frame-wise encoder and decoder, transformer decoder. These modules can be implemented with any network. However, in our paper, as acknowledged by the reviewer, we employ existing frameworks to achieve these functional modules. This approach allows us to: 1) fairly compare with previous methods by sharing the same backbone, and 2) verify the effectiveness of our pipeline, which relies on the new paradigm rather than network design.
>
> Furthermore, although these existing frameworks have been successful in previous works, they may not be effective in our new pipeline, which is organized with different functional modules.  Therefore, in terms of TAS tasks, we have made numerous efforts to optimize our approach. These efforts include experimenting with different matching strategies, establishing connections between frame decoders and transformer decoders, and implementing auxiliary loss functions. We analyze these aspects in detail in the main paper and supplementary materials and hence provide insights for designing a superior pipeline.
>
> It is worth mentioning that there are many impactful works to adopt the DETR-style model into different fields as what we do in TAS tasks, e.g., TrackFormer[1] for multi-object tracking, MaskFormer[2] for image segmentation and CLTR[3] for crowd localization. However, our work is the first such exploration for TAS.
>
> [1] T. Meinhardt, A. Kirillov, L. Leal-Taixe, and C. Feichtenhofer. Trackformer: Multi-objecttracking with transformers. In Proceedings of the IEEE/CVF conference on computer vision and pattern recognition, pages 8844–8854, 2022.
>
> [2] B. Cheng, A. Schwing, and A. Kirillov. Per-pixel classification is not all you need for semantic segmentation. Advances in Neural Information Processing Systems, 34:17864–17875, 2021.
>
> [3] D. Liang, W. Xu, and X. Bai. An end-to-end transformer model for crowd localization. In European Conference on Computer Vision, pages 38–54. Springer, 2022.
>
> **2 .For Weaknees 2**
>
> (**Running Time**) Thank you for the suggestion to include detailed results to improve our paper. Based on the reviewer's suggestion, we have added the table below. And find that the original module's additional running time is due to the NMS post-processing. To ensure a fair comparison, we have listed both the forward time and the post-processing time below. Thanks again for the reminder and we will update the manuscript.
>
> |Method | Total time(s) | Backbone time(s) | Post-processing time(s) |
> |:--------:|:----------------:|:----:|:--------:|
> |SSTCN| 0.080 |0.031 | 0.049|
> |BaFormer(SSTCN) | 0.074 | 0.040 | 0.034|
> |ASFormer(Encoder) | 0.158 |0.092 | 0.066 |
> |BaFormer(ASFormer Encoder) |0.139 |0.105 | 0.034 |

---

> > ### Comment · Reviewer_KTbs · 2024-08-12
> > **Response to the authors.**
> >
> > Thanks the authors for the reply, my concerns, especially the fair comparison part are mostly solved. I will keep my current rating as borderline accept, slightly towards 6 due to the novelty issue.

---

> > > ### Author Response · Authors · 2024-08-12
> > >
> > > Thank you very much for your time and effort in reviewing our work. I would appreciate the opportunity to further discuss and address your concerns regarding the novelty of our approach.

---

### Official Review · Reviewer_KZs3 · 2024-07-12

**Soundness:** 3
**Presentation:** 3
**Contribution:** 3
**Rating:** 6
**Confidence:** 4

**Summary:**

The authors introduce BaFormer, an innovative single-stage boundary-aware Transformer network designed for temporal action segmentation (TAS). This method employs instance queries for instance segmentation and a global query for class-agnostic boundary prediction, achieving substantial reductions in computational costs while preserving or enhancing accuracy relative to state-of-the-art approaches. Experiments across multiple datasets highlight BaFormer's efficiency and effectiveness, demonstrating its capability to achieve competitive results with significantly lower running time and resource usage.

**Strengths:**

1. The proposed approach is interesting and show good performances on various of datasets.


2. The method is clearly described and easy to understand. The whole paper is well-written,


3. Efficient temporal action segmentation is an important research direction.

**Weaknesses:**

1. Lack of the ablation studies. The authors are encouraged to add more ablation studies, e.g., regarding different number of query tokens. Apart from those ablations, the ablation of the loss function is also interesting since there are so many losses incorporated in the proposed method.


2. Lack of qualitative results.  The authors are also encouraged to provide some qualitative results or TSNE figure to show more insights regarding the efficacy of the proposed method compared with some baselines.


3. Since the authors mentioned that the proposed method is efficient, however when we have a look at Table 1, the proposed approach do not clearly show the benefits in terms of the model size compared with DiffAct, which is regarded as one weakness.

4.  The sensitivity of the parameters is interesting to be analyzed, since the proposed method do preserve large amount of the hyper parameters.

**Questions:**

1. How could the variation in the number of query tokens impact the results of the proposed method?


2. Can the authors provide a detailed comparison of the different loss functions used in the proposed method?

3. Could you provide examples of baseline methods that should be included in the qualitative comparison?

4. Can you provide examples of how changes in hyperparameters could affect the performance of the proposed method?

**Limitations:**

No, the authors mentioned that the limitations are included in the supplementary however there is no supplementary materials for this work.

---

> ### Author Rebuttal · Authors · 2024-08-07
>
> Thank you for the valuable suggestions to enrich the analysis of our model, most of which are included in the existing supplementary. It looks like you didn’t find the supplementary material -- we double-checked and it’s not a problem, and other reviewers haven’t had this issue either. Please inform us or contact the conference organizer/AC if there are any issues with locating or downloading the supplementary material. We refer to the location of the following question in our supplementary material.
>
> (**Weakness 1 and Question 1: more ablation studies and query tokens**)  We agree that more ablation studies help us to analyze the model further, and we include some of them in supplementary,  such as the number of query tokens, single/multi-level feature connections, auxiliary loss, and
> other factors.  Specifically, as for the number of query tokens, the experiment indicates that a slight increase in the number of queries can enhance the model’s ability without significantly raising computational costs. However, the model’s performance peaks at 100 queries, achieving the highest accuracy of 89.5% at a computational cost of 4.45G FLOPs (more details can refer to Table 2 of the supplementary material).
>
>
> (**Weakness 1 and Question 2: a detailed comparison of the different loss functions**) In supplementary Table 3, we present the performance results when the auxiliary loss is included under single or multiple features. Notably, applying auxiliary losses in multi-level features yields a substantial improvement in accuracy (2.1%), surpassing the 0.5% accuracy gain observed in single-level features. This suggests that auxiliary losses have a more pronounced effect on multi-level feature representations.
>
> (**Weakness 2 and Question 3: qualitative results**) Qualitative results, which can help us to analyze segmentation results, especially among the time dimension. So, in the supplementary material, we also provide additional qualitative results in Figure 2 to demonstrate the efficacy of our proposed query voting.  In the 50Salads dataset, we present enhanced visualizations for a more comprehensive analysis. The figure includes the outcomes of instance segmentation, the segmentation results without boundary integration,  and the ground truth.
>
> (**Weakness 3: model size compared with DiffAct**)We apologize for the concerns and recognize that we should clarify that efficiency refers to shorter running time. Thus, despite DiffAct's smaller size, its running time is longer than ours. DiffAct is a multi-stage method based on the diffusion model. It is built on a model with a small size but requires 25 iterations to refine the results, significantly increasing the running time. By contrast, our BaFormer requires only 6% of the time used by DiffAct and obtains a comparable performance. Directly comparing single-stage methods with multi-stage ones is somewhat unfair, we therefore divide them into two groups in Table 6. Alternatively, we can adjust the models to have similar running times for a fair comparison, we present the results in Table 7. Our method shows superior performance, achieving an accuracy of 89.5% and F1@10 of 89.3, compared to DiffAct's accuracy of 74.2% and F1@10 of 48.3 (more experimental results with different steps for DiffAct are shown in Table 4 in the supplementary material). Furthermore, we can extend our method to a multi-stage one to increase the accuracy to 90.5%, which surpasses the original DiffAct. However, in this paper, we focus on improving the performance of single-stage methods rather than relying on multi-stage methods.
>
> (**Weakness 4 and Question 4: more analysis of hyperparameters**) The supplementary material includes an analysis of hyperparameters, such as the number of the query tokens(Table 2), the connection between the frame-wise and transformer module (Table 3),  the transformer decoder layer(Table 1),  the single or multiple features (Table 3) and so on.
>
> (**Limitation**) The limitation is also discussed in the supplementary. Our proposed approach is under the query-based Transformer framework, which is well-known for its slow training convergence and data thirty. We observe that training query-based Transformer converges slower than training frame-wise approaches. For the discontinuous binary mask predictions, we assume that it may also be attributed to the limited data of action segmentation benchmarks. We hope that these factors can inspire future work.

---

> > ### Comment · Reviewer_KZs3 · 2024-08-09
> > **Response to the authors**
> >
> > Thank you for your response.
> >
> > However, according to the instructions for the paper submission, the technical appendix should be in the main submission pdf together with your main paper and checklist. Other supplementary materials such as data and code can be uploaded as a ZIP file. But it is ok for me if it does not violate the review policy.
> >
> > Could you provide more detailed analysis regarding why auxiliary losses work better on multi-level features compared with single-level features?

---

> > > ### Author Response · Authors · 2024-08-09
> > > **More analysis auxiliary loss on multi/single-level features**
> > >
> > > Thank you again for your valuable feedback and comments. We will put the technical appendix together in the manuscript.
> > >
> > > (**auxiliary losses work better on multi-level features compared with single-level features**).
> > > We explain how the auxiliary losses work to address your concern about why they work better on multi-level features than single-level features.  The auxiliary losses originate from each Transformer decoder layer, excluding the final layer. According to the formulation of outputs from each Transformer decoder layer (The definitions of variables are provided in the paper.):
> > >
> > > $\textbf{X}_i=$
> > >
> > > $\text{Linear}(\text{softmax}(\frac{\textbf{P}^m_{i-1}}{\sqrt{C}}\odot\hat{\textbf{Q}}_i\hat{\textbf{K}}_i^T)\hat{\textbf{V}}_i)+$
> > >
> > > $\textbf{Q}_{i-1}$
> > >
> > > , the difference in auxiliary losses between single-level and multi-level features lies in the origin of $\hat{\textbf{K}}_i$ and $\hat{\textbf{V}}_i$. Specifically, as illustrated in Figure 1 of the supplementary material about the details of connections, in the single-level feature connection, $\hat{\textbf{K}}_i$ and $\hat{\textbf{V}}_i$ are derived from the outputs of the frame-wise encoder $\textbf{F}_e$ and remain consistent across different Transformer decoder layers. In contrast, in the multi-level feature connection, $\hat{\textbf{K}}_i$ and $\hat{\textbf{V}}_i$ are obtained from the outputs of different frame-wise decoder layers $\textbf{f}_i$, as shown in Equation 2 of the paper. According to backpropagation theory, the frame-wise decoder can undergo more gradient updates in the multi-level feature connection. Moreover, as outlined in Equation 3 and 4 of the paper, since the frame-wise decoder contributes to both query mask and boundary prediction, enhanced parameter learning in this component benefits the entire model.

---

> > > > ### Comment · Reviewer_KZs3 · 2024-08-11
> > > > **Response to the authors**
> > > >
> > > > Thanks the authors for the reply, my concerns are mostly solved. I would like to improve my rating to 6.

---

> > > > > ### Author Response · Authors · 2024-08-12
> > > > > **Thank you!**
> > > > >
> > > > > We sincerely thank your constructive suggestions and questions to enhance our paper.

---

### Official Review · Reviewer_W81q · 2024-07-13

**Soundness:** 2
**Presentation:** 2
**Contribution:** 3
**Rating:** 4
**Confidence:** 3

**Summary:**

This paper proposes a fully-supervised TAS approach that predicts both segment and frame predictions. Two levels of predictions are then combined to form the final action boundaries. The approach achieves competitive performance and at the same time reduces inference time for better effiency.

**Strengths:**

1. The motivation of leveraging segment-wise prediction to help reduce the temproal dimension processed by model is sound. The design drawing inspiration form image classification looks soild for TAS.

2.  The apporach successfully boosted the processing efficiency of action segmentation while maintaining high accuracy on common TAS benchmarks.

**Weaknesses:**

The reviewer understands that the authors' main objective claimed is to reduce the processing time. However, given the design of voting between frame-wise and segment-wise predictions, one expected advantage of having access to segment prediction is to reduce the over-segmentation issue, yet the effect of such voting scheme between frame-wise and segment-wise predictions did not achieve the expected effect. On most of the metrics, the propose approach did not manage to outperform existing SOTA approaches, for example DiffAct, especooally on the segmental metrics.

The paper is diffult to parse from Sec 3 with very confuing notations.  For example:

1. L.102 frame-wise encoder-decoder, the encoder is the feature extractor and what is the decoder? why is it called a decoder?Also what is the value for dim $C$ used in exps? The reviewer failed to find such information.

2. Likewise, L. 107 $P_i^m = \phi_m(Q_l, F_d)$, what does $m$ indicate? and why the underscripts are inconsistent (m, l, d) in the equation, and it makes the reviewer guess about the meaning. The definitions of variable shall always be around where the variables are first introduced in text.

The authors should consider explain these notations for example with all variables at the same layer, i.e., a fixed underscript for all varibles to faciliate a clear understanding.


The current rating is boardline reject given the above concerns but the reviewer would consider raise the rating if the authors could address the issues.

**Questions:**

1. What is the advantage of leveraging an extra branch of segment predict? What advantage did it bring in terms of accuracy and combating the issue of over-segementation?

2. Clarifications reagarding the confusing points listed in weakness.

**Limitations:**

The authors had discussed in the supp.

---

> ### Author Rebuttal · Authors · 2024-08-07
>
> We thank the reviewer for acknowledging our contributions and below is our response regarding the reviewer’s concerns.
>
> **1.For Weakness**:
> >However, given the design of voting between frame-wise and segment-wise predictions, one expected advantage of having access to segment prediction is to reduce the over-segmentation issue, yet the effect of such voting scheme between frame-wise and segment-wise predictions did not achieve the expected effect. On most of the metrics, the propose approach did not manage to outperform existing SOTA approaches, for example DiffAct, especooally on the segmental metrics.
>
> **Response**:
>
>  We apologize for not providing a clear note on how to compare our performance with others, despite having separated the methods into single-stage and multi-stage in Table 6. Since our work focuses on the efficiency aspect of TAS, it is somewhat unfair to compare the single-stage (i.e., BaFormer) and multi-stage (i.e., DiffAct) models on segmental metrics. That is why we categorize all methods into multi-stage and single-stage groups, as shown in Table 6. Specifically, multi-stage models involve many stages of refinement, which are time-consuming. Our single-stage model is designed to be more efficient. Although our model shows a 0.5% decrease in F1@10, we improve accuracy by 0.8% on the 50Salads dataset, and notably, our model requires only **6%** of the inference time compared to DiffAct. DiffAct, based on a diffusion model, includes 25 inference steps, significantly increasing the time required to refine segmental metrics. When comparing models with similar inference times, DiffAct achieves **64.9%** on F1@10 and **88.6%** accuracy, while our method achieves **89.3%** on F1@10 and **89.5%** accuracy, demonstrating superior segmental metrics of our model. Further details can be found in Table 11 of the supplementary.
>
> We can extend our methods to multi-stage to improve the segmental metrics further; however, experimental results indicate that this increases processing time, which contradicts our primary objective in this paper—to design an efficient TAS model, as acknowledged by the reviewer. To demonstrate the effect of our model, we also extended it into a multi-stage framework and modified the visual features for accuracy 91.2% and 90.8%, which are superior to DiffAct.
>
> To fairly assess the effectiveness of the voting mechanism in reducing the over-segmentation issue, we need to compare it within single-stage models. These models all include one stage with post-processing, and comparisons should focus on different processing techniques with similar backbones. For instance, compared to UVAST with FIFA, our method demonstrates superior performance in segmental metrics (88.9% vs. 88.3%), accuracy (89.5% vs. 84.5%), and faster inference (0.139s vs. 1.765s), which result from the query-based model design. **This demonstrates that our approach has already outperformed existing SOTA approaches for the one-stage model.**
>
> **2. For weakness of confusing notation in Sec 3**
>
> (**About decoder**) Sorry for the confusion. **It should be noted that the encoder is frozen in all previous work for fair comparison**, mentioned in the experiment of the paper. Thus, the decoder aims to further process features from the encoder into high-level information, making them more adaptable to the TAS task through training. We call it a decoder to distinguish it from the frozen feature encoder/extractor. The decoder architecture can be any network adopting from the existing work such as MSTCN and Asformer.
>
> ($C$ **in exps**) In the model, all instances of $C$ are set to 64, which is the feature dimension shared by different features. The value can be found in L217 of the experiment settings with "an output dimension of 64". We will clarify this by changing the phrase to "an output dimension $C$ of 64".
>
> (**Notation clarity**) Thanks for the suggestion and we will add improve the clarification and eliminate the notation inconsistency in the revision. Specifically, $\( \textbf{P}_i^m = \varphi_m(\textbf{Q}_l, \textbf{F}_d) \) $ represents the \(i\)-th mask features. Here, $m $ is the abbreviation of "mask".
>
> We have reviewed all the notations and will use superscripts to represent the type of the variable and subscripts to represent the index. We will update the following notations:
>
> |        Old  |  New | Meaning |
> |:--------:|:----------------:|:----:|
>  | $ \textbf{F}_e, \textbf{F}_d   $  |  $ \textbf{F}^e, \textbf{F}^d $| the output features of **e**ncoder or **d**ecoder |
>  |  $ \varphi_c, \varphi_m,\varphi_b $ |  $ \varphi^c, \varphi^m,\varphi^b $ | heads for predicting query **c**lass, query **m**ask, **b**oundary  |
>
> Furthermore, for better understanding existing notations, we will add the following descriptions:
> |     Notation  |  Meaning |
> |:--------:|:----------------:|
> |$ \textbf{Q}_l $ | instance query embedding from $l^{\text{th}}$ layer |
> | $ \textbf{P}_i^m $| Query masks $\textbf{P}^m$ from $i^{\text{th}}$ layer (the $m$ represents **m**ask.) |
> | $\textbf{P}_i^b $| Boundary probability $\textbf{P}^b$ from $i^{\text{th}}$ layer (the $b$ represent **b**oundary.)  |
> | $\textbf{y}_i^m $| binary mask $\textbf{y}^m$ from the $i^{\text{th}}$ query (the m represents **m**ask) |
>
> **3. For Question 1**
>
> (**Advantage of extra branch**). As our approach is based on voting to yield segmentation results, the extra branch for boundary generation is essential for our method. The extra branch can generate high-quality boundaries without much additional effort (nearly 5% Flops assumption). It can bring a 2.3% improvement in accuracy and a 2.1% improvement in Edit score, compared to a direct summary of all query predictions without an extra boundary prediction branch.
>
> **4. For Question 2**
>
> Refer to the answer of above part "2. For weakness of confusing notation in Sec 3" .

---

> > ### Comment · Reviewer_W81q · 2024-08-13
> >
> > The reviewer thanks the authors for providing the detailed rebuttal.
> >
> > The intent behind the first raised weakness was to request an explanation or analysis regarding why the combination of frame-wise and segment-wise prediction underperforms in the segmental metrics. Could this issue be attributed a weaker backbone?The author could also consider including an ablation study to show where the performance gain come form, i.e., starting query based segmentation to evaluate how well the query-based instance segmentation works and  then add in the boundary branch to show its effectiveness. The reviewer assumes that this corresponds the authors reponse to Q1? It would be much appreciated if the authors could provide a detailed table containing such information.
> >
> > The second notation table is wrong? Should it be $p_i^m$ instead of $P_i^m$? and similarly $P_i^b$. What is the relationship about $l$ and $i$?  are they identical?
> >
> > As a side comment, why are the main comparsion constrained to the single stage approaches only? The main objective is the model efficiency, so as long as the multiple stage approaches can reach a similar running time, they are good counterparts for comparsion.

---

> ### Author Response · Authors · 2024-08-13
> **Detailed table and Notation**
>
> We thank the reviewer for providing us with the opportunity to conduct a more detailed analysis of the performance on segmental metrics.
>
> **1.Respond to detailed table and gains**
>
> (**Detailed table** ) As the reviewer suggested, the answer is "Yes"—the weaker backbone,  which we adopt for fair comparison, is one of the reasons for the underperformance in segmental metrics. Therefore, the detailed results of adopting another visual encoder, TSM[1], in the backbone to achieve superior performance are as follows:
>
> |   Visual Encoder  |   F1@{10,20,50}  | Edit | Accuracy |
> |:--------:|:----------------:|:----:|:--------:|
> |   I3D  | 89.3  88.4   83.9 | 84.2|89.5 |
> | TSM[1] | 90.7   89.7    85.6 | 86.0|  90.8|
>
> This demonstrates that a stronger backbone can improve segmental metrics, with F1@10 increasing by 1.4%, Edit by 1.8%, compared to a weaker backbone.
>
> [1] J. Lin, C. Gan, and S. Han. Tsm: Temporal shift module for efficient video understanding. In Proceedings of the IEEE International Conference on Computer Vision, 2019
>
>
> (**Performance gains**)The answer is "Yes" to the reviewer's assumption of the performance gains corresponds to the above response to Q1. To provide more details on where the performance gains come from, the reviewer's suggestion to present the performance of query-based instance segmentation, both with and without the addition of the boundary branch, is highly valuable. The results are as follows:
>
> |    Method     |query-based instance segmentation | boundary branch    | F1@{10,20,50}  | Edit | Accuracy |
> |:--------:|:----------------:|:----:|:--------:|:--------:|:--------:|
> | A         | ✓	        | 	    |86.5    85.9    80.6  | 82.1 | 87.2|
> |B           |   ✓	        | 	✓ | 89.3    88.4    83.9  | 84.2 | 89.5|
>
>
> **2. Respond to the notation**
>
> (**Notation**) We appreciate the reviewer's efforts to help us in clarification. The second notation table is correct but little confusing; both the $\textbf{p}^m$ and $\textbf{P}^m$ exist, and their relationship is described in L156 of paper, $\textbf{P}^m = (\textbf{p}^m_i)_{i=1}^M$. To clarify, we assign the $l$ as the index for the layer of the transformer decoder, while the $i$ is the index for the query. Then the above notation table can be modified to:
>
> |     Notation        |Meaning |
> |:--------:|:----------------:|
> |$\textbf{Q}_l$ | instance query embedding from the $l^{th}$ layer|
> |$\textbf{P}^m_l$ |Query masks prediction from the $l^{th}$ layer (the $m$ represents **m**ask.)|
> |$\textbf{p}_i^m$|The $i^{th}$ query mask prediction in $\textbf{P}^m$ ( the index $l$ is omitted in $\textbf{P}^m$ for simplicity )|
> |$\textbf{y}_i^m$| binary mask from the $i^{th}$ query (the $m$ represents **m**ask.) |
>
> **3. Respond to comparison**
>
> (**comparsion**)In the paper, shown in figure 1, most of two-stage and one stage methods cannot achieve a good trade off between effectiveness and efficiency. Thus, we categorize all the methods into two classes to see what achievement we have for temporal action segmentation.
> In addition to Table 6, we also consider recent multi-stage methods, adjust them to have similar running times, and present the results in Table 7. These results demonstrate that we achieve better performance when our method operates with similar time constraints to the multi-stage methods. Furthermore, a more detailed comparison with multi-stage methods across different numbers of stages is included in Table 4 of the supplementary material.

---

### Official Review · Reviewer_rrtF · 2024-07-13

**Soundness:** 3
**Presentation:** 3
**Contribution:** 3
**Rating:** 4
**Confidence:** 4

**Summary:**

The paper introduces BaFormer, a boundary-aware Transformer network designed to enhance the efficiency of Temporal Action Segmentation while maintaining high performance. BaFormer tokenizes each video segment as an instance token for intrinsic instance segmentation and employs instance queries for segmentation and a global query for class-agnostic boundary prediction. This approach allows for continuous segment proposals and a simple voting strategy during inference.

**Strengths:**

1, The proposed Query Voting method, though simple, is highly effective as demonstrated by the ablation experiments in Table 3.

2, The proposed BaFormer being a single-stage method is a practical advantage.

3, The proposed BaFormer achieves performance close to state-of-the-art two-stage methods.

**Weaknesses:**

1, One of the major innovations, Boundary-aware Query Voting, is implemented only during the inference stage, and its process is very simple. The method involves summing up query masks within segment proposals and selecting the highest, which is a common approach in many temporal action proposal works. This raises doubts about the innovation's originality. Additionally, BaFormer's overall network design follows previous works, with the main innovations being in the Matching Strategies and Voting.

2, Besides I3D, the authors should compare the impact of other visual encoders on the model's performance. Understanding the effects of different visual features is crucial, and the visual encoder used in Table 6 should be clearly stated to ensure fair comparisons.

3, In Table 7, the terms "CNN and Transformer backbones" need clarification. The results for BaFormer with a Transformer-Base backbone correspond to those in Table 6, suggesting that "backbone" might refer to the Frame Decoder. This is inconsistent with the "Frame-wise Encoder-Decoder" terminology used by the authors.

4, The authors incorrectly cite KARI[18] as a NeurIPS 2024 paper, whereas it is actually from NeurIPS 2023. It seems the authors are not aware that they are submitting to NeurIPS 2024 themselves :)

5, Figure 6 aims to explain why Query Voting is superior to frame-based voting, but the visualization of queries and the meaning of different colors are not clearly explained, causing confusion for readers. Additionally, it would be interesting to know the performance of frame-based methods on shorter segments, as they might perform better than query-based methods in such cases.

**Questions:**

The questions have been detailed in the "Weaknesses".

---

> ### Author Rebuttal · Authors · 2024-08-07
>
> Thank you for your comments. We provide discussions and explanations about your concerns as follows.
>
> **1. For Weakness1**
>
> (**Query voting is very simple**) We thank you for acknowledging our contributions to Matching Strategies and Voting. Regarding your concern about the simplicity of the process, we intentionally designed it to be simple to contribute to efficiency.  In our paper, we aim to design an effective TAS model with high efficiency. So when designing such a model, we pursue two key points: (1) a highly efficient model structure to avoid heavy multi-stage or multi-step architecture; (2) integration of simple but effective post-processing upon high-quality model predictions. Specifically, our model transforms frame-wise predictions into query class-mask predictions for TAS. At the same time, we put much effort into exploring different matching strategies and extensively ablate different components (number of decoder layers, query quantity, auxiliary loss, global query for boundary generation, etc., as reported in experiments and supplementary material.) of the network architecture. **After achieving high-quality query class-mask predictions as well as precise boundary generation, we are then enabled to use very simple but highly efficient query voting to realize the expected efficient TAS**.
>
> We can also adopt more complicated post-processing, such as the Viterbi algorithm, activity grammar, and so on. However, previous papers have verified that they are time-consuming, which is not what we desired. For example, in another single-stage method UVAST, when using the Viterbi algorithm it spends more than **480s** with an accuracy of **87.4%** on 50Salads dataset; while when using simpler post-processing, i.e., alignment decoder, it takes **0.577s** but with a degraded accuracy of **79.5%**. By contrast, our BaFormer employs simple voting with **0.139s** and achieves a higher accuracy of **89.5%**.
>
> (**BaFormer follows previous work**) As for the concern that BaFormer follows previous work, BaFormer is designed to improve efficiency, and we experientially find that a query-based model can change the dense information into sparse information to reduce FLOPs. So, we follow the DETR-style pipeline to predict the query-based results. It gives us insight to take a new paradigm in TAS. **Our focus is on how to innovatively adopt the DETR-style model into TAS that has not been fully explored by such an approach**. Like BaFormer, many other promising works also follow the DETR model to achieve their different targets, for example, MaskFormer in image segmentation, CLTR  in crowd localization, TrackFormer  in multi-object tracking, and so on. They design new pipelines similar to DETR for different tasks as we do.
>
>  **2. For Weakness 2**
>
> (**Other visual encoder**) Thank you for the suggestion to adopt other visual encoders, which will enrich our model's performance. Accordingly, we conducted experiments with TSM visual encoders compared with I3D. The results on 50Salads are as follows:
>
> |    Visual Encoder      |   F1@{10,20,50}  | Edit | Accuracy |
> |:--------:|:----------------:|:----:|:--------:|
> |    I3D | 89.3 | 88.4 | 83.9 | 84.2 | 89.5 |
> |    TSM | 90.7 | 89.7 |  85.6 | 86.0 | 90.8|
>
> The results indicate that a superior visual encoder can further enhance performance.
>
>  **3. For Weakness 3**
>
> (**Terminology**) Sorry for the confusion. In our study, the term "CNN backbones" refers to the single-stage MSTCN, and the "Transformer backbone" refers to the single stage of ASFormer. In "Frame-wise Encoder-Decoder", the structure of the frame-wise encoder is fixed as I3D for all methods in Table 7. Therefore we focus on different structures of the frame-wise decoder. As acknowledged by the reviewer, the term "backbone" specifically refers to the frame-wise decoder. We apologize for any confusion and thank the reviewer for pointing it out. For clarity, we have replaced the term "backbones" with "frame-wise decoder".
>
>  **4. For Weakness 4**
>
> (**Typo in reference**) Thank you for pointing out this embarrassing typo. We promise to fix it and check carefully other references in the revision.
>
>  **5. For Weakness 5**
>
> (**Visualization in Figure 6**) Again, sorry for the confusion, and we will add relevant clarification in the revision. Figure 6 comprises two parts. Specifically, to generate the top part, we binarize the predicted masks $\in R^{n\times T}$ to 0 or 1 with a threshold 0.5, where $n$ is the number of queries, and $T$ is the length of the video. Thus, the positions with a value of 1 indicate the presence of actions corresponding to the query class. Next, we apply different colors to these positions and each color represents a single action. Then query masks are stacked to form the top part. In the bottom part, we present video segment results of frame-based and query-based voting, along with the ground truth. The red arrow points towards a specific query within a segment proposal, illustrated between two vertical black dashed lines. The red dashed box shows the segment results within the proposal.
>
>
> (**Performance on segments with different lengths**) Thanks for the suggestion. We conducted experiments with different segment lengths and the results are summarized below. We find that the performance of frame-based methods is slightly better on shorter segments, and the accuracy is shown below:
>
>
> |  Length   |0~1000 | 1000~2000 | 2000~ 3100 |
> |:--------:|:----------------:|:----:|:--------:|
> |    Frame-based | 87.68 | 90.98 | 92.47|
> |    Query-based | 86.85 | 92.01 | 95.38|

---

> > ### Comment · Reviewer_rrtF · 2024-08-13
> > **Response to the authors**
> >
> > Thank the authors for their detailed responses to the review.
> >
> > Regarding the response to Question 1, while I understand the importance of maintaining concise perspectives in both the model and post-processing stages, I still hold reservations about the novelty of the proposed method.
> >
> > Concerning Question 2, I am puzzled by the decision to select I3D as the backbone, given that TSM outperforms I3D in the results.
> >
> > Questions 3 to 4 have been thoroughly addressed.
> >
> > Regarding Question 5, the performance of frame-based methods on shorter segments is better than that of query-based methods. This is a significant drawback of the proposed query-based method that also needs to be considered.
> >
> > Finally, I remain concerned about the modest performance improvement compared to the 2022 UVAST method on GTEA.

---

> ### Author Response · Authors · 2024-08-13
>
> Thank you for the valuable feedback and the opportunity for further discussion.
>
> **1.Respond to I3D or TSM**
>
> (**I3D or TSM**) We understand the reviewer's concerns regarding the results under different backbones. To ensure a fair comparison, all methods in Table 6 of the paper use I3D features. Specifically, for all methods, TAS tasks use fixed I3D features rather than TSM features as input. We believe it is fairer to use I3D as the visual encoder, even though TSM features may yield superior performance.
>
> **2. Respond to shorter segments**
>
> (**For shorter segments**) Thank you for the valuable suggestion, which has helped us analyze methods more deeply across different segment lengths. We will include additional analysis based on the table presented for Question 5. This type of case arises from mask prediction in the query-based method. In binary mask prediction for shorter segments, the ratio between foreground and background is smaller than in longer segments, leading to varying accuracy in mask prediction across different segment lengths. This is similar to the challenge faced in small object detection using DETR. We will include this observation in the limitations of the manuscript.
>
> To improve this case, potential solutions could involve reweighting the ratio of positive to negative samples, increasing the weight assigned to the mask prediction heads, and so on. This approach could guide our efforts in future work.
>
> **3.Performance on GTEA**
>
> (**GTEA**)We acknowledge the reviewer's concern regarding the performance on GTEA. While our method results in a 0.7% decrease in F1@10, it improves accuracy by 3%. This outcome is likely related to the shorter segment lengths in GTEA as described in part “for shorter segments”. We have provided data statistics across different datasets, where "Ratio" refers to the ratio between segment length and video length.
>
> |Dataset|number of Video| video length(min)|segment length(s)| Ratio|
> |:--------:|:--------:|:--------:|:--------:|:--------:|
> |GTEA |28 |1.24 | 2.21 | 0.0297|
> |50Salads|50|6.4|36.8|0.0958|
> |Breakfast|1712| 2.3|15.1|0.1094|
>
> We observe that the ratio in GTEA is only 1/3 of that in 50Salads or Breakfast. Beyond accuracy, for efficiency, we achieve a running time of approximately 1/5000 of UVAST's time, resulting in a significantly faster TAS.

---

### Decision · Program_Chairs · 2024-09-25

**Decision:**

Accept (poster)

**Comment:**

This paper received mixed reviews, with two borderline rejects, one borderline accept and one accept.  All the reviewers appreciate the methodology and effectiveness, while there are some concerns with respect to technical clarifications.

The AC has carefully reviewed the paper, author response and discussion and sees value in the work to the NeurIPS community and therefore recommends acceptance.  The authors are recommended to revise their work based on the discussions and clarifications with the reviewers.